# *MCL-1* gains occur with high frequency in lung adenocarcinoma and can be targeted therapeutically

Enkhtsetseg Munkhbaatar ⬤ et al.#

Evasion of programmed cell death represents a critical form of oncogene addiction in cancer cells. Understanding the molecular mechanisms underpinning cancer cell survival despite the oncogenic stress could provide a molecular basis for potential therapeutic interventions. Here we explore the role of pro-survival genes in cancer cell integrity during clonal evolution in non-small cell lung cancer (NSCLC). We identify gains of *MCL-1* at high frequency in multiple independent NSCLC cohorts, occurring both clonally and subclonally. Clonal loss of functional *TP53* is significantly associated with subclonal gains of *MCL-1*. In mice, tumour progression is delayed upon pharmacologic or genetic inhibition of MCL-1. These findings reveal that *MCL-1* gains occur with high frequency in lung adenocarcinoma and can be targeted therapeutically.

---

#A list of authors and their affiliations appears at the end of the paper.

                                                                    

Lung cancer is the most commonly diagnosed cancer and the leading cause of cancer-related deaths worldwide[1]. Despite the identification of genotype-directed targeted therapies and the use of immune checkpoint inhibitors, lung cancer is still associated with substantial morbidity and mortality[1].

Based on its histology, lung cancer is divided into small cell lung cancer (SCLC) and non-small-cell lung cancer (NSCLC), with the latter representing about 85% of total cases. Lung adenocarcinoma (LUAD) and lung squamous cell carcinoma (LUSC) are the most common subtypes of NSCLC[2].

Approximately 33% of adenocarcinoma harbour activating mutations in KRAS[3], which remain clinically difficult to target. A novel class of inhibitors against mutant KRAS has just started to be tested in clinical trials[4,5], while targeting signalling elements downstream of KRAS, such as MAPK, has yielded unsatisfactory results in clinical trials[6,7]. Therefore, patients with KRAS-mutant lung cancer often relapse indicating a substantial unmet medical need for these patients[2,4]. Loss of the TP53 tumour suppressor is another common event in lung cancer[3]. Loss of TP53 function has been observed in 46% of lung cancer patients and is often detected in combination with additional mutations in common oncogenic drivers, such as gain-of-function mutations in the epithelial growth factor receptor, EGFR[3]. TP53 protects cells from tumorigenesis by activating a variety of anti-proliferative and stress-response pathways, including apoptosis, senescence, cell cycle arrest, and the coordination of DNA damage repair[8,9].

Protection against programmed cell death pathways, like apoptosis, represents a critical form of oncogene addiction in cancer cells: the substantial cellular stress that cancer cells experience during their transformation and uncontrolled proliferation requires a reliance on signalling pathways that secure their continued survival[10].

Apoptosis is largely regulated by a balance between pro-survival and pro-apoptotic members of the B cell lymphoma 2 (BCL-2) protein family. Accordingly, pharmacologic inhibitors targeting pro-survival BCL-2 proteins have been developed to induce apoptosis in cancer cells[11]. Various malignancies show recurrent genetic amplifications affecting pro-survival members of this family. For example, analysis of somatic copy number alterations of different human cancer specimens and cell lines revealed genomic amplifications of the chromosomal region containing the anti-apoptotic gene Myeloid Cell Leukaemia 1 (MCL-1) in various cancer types[12,13]. Antisense oligonucleotides targeting MCL-1 were able to induce apoptosis in certain NSCLC cell lines, suggesting a functional relevance for MCL-1 in NSCLC survival[14,15]. However, little is known about the role of MCL-1 during lung cancer clonal evolution and maintenance.

We provide evidence that the selective pressure for the evasion of apoptosis during tumour evolution results in recurrent genomic gains of MCL-1 with high frequency in NSCLC patients. Functionally, we demonstrate that Mcl-1 deletion impairs the development of Kras-driven tumour and that pharmacologic inhibition of MCL-1 in fully established Kras-mutated p53-deleted murine lung cancers slows tumour progression. Together, these data identify MCL-1 as a rational target for the treatment of NSCLC patients.

## Results

### Genomic and protein gains of MCL-1 in lung cancer. To understand the preponderance of genomic gains of the pro-survival gene MCL-1 in cancer, we analysed genomic data of 24 different cancer entities from The Cancer Genome Atlas (TCGA). In line with previous data[12,13], MCL-1 was significantly gained and highly expressed in 22 cancer types (Fig. 1a).

We further analysed the TCGA dataset for LUAD and LUSC in order to specifically understand the role of MCL-1 in lung cancer. Analysis of LUAD samples revealed MCL-1 gains in 417 out of 652 LUAD samples (64%), of which 70/417 (11%) were high-level gains ($>2 \times$ ploidy, Fig. 1b). We detected a positive correlation between MCL-1 genomic gain and mRNA levels (Fig. 1c), suggesting a functional significance of the genomic gains. Analysis of LUSC samples showed MCL-1 gains in 211 out of 652 samples (32%), of which 28/211 (4%) were high-level (Fig. 1d). Although MCL-1 was less frequently gained in LUSC, we still identified a consistent correlation with its mRNA levels (Fig. 1e).

To further dissect the impact of MCL-1 in lung cancer, we took advantage of the TRACERx study[16] (patient information in Supplementary Fig. 1). Analysis of LUAD samples revealed MCL-1 copy number gains in 46/61 (75%) tumours, of which 10/46 (22%) were high-level (Fig. 1f). Similar results were detected in LUSC tumours, 19/32 (59%) tumours showed MCL-1 gains and 1/19 (5%) was classified as a high-level gain (Fig. 1g). MCL-1 maps to the human chromosome 1q21, a region that is frequently altered in pre-neoplastic lesions and malignant diseases[17]. Analysis of the size of genomic segments affected by copy number alterations revealed that MCL-1 is subject to both focal and broad level gains, with focal events (<20 Mb) exhibiting a higher preponderance of high-level gains compared to broad ones (Supplementary Fig. 2).

The correlation between genomic gain and mRNA levels was confirmed in these cohorts (Fig. 1h, i). A similar relationship was observed for additional 316 genes for LUAD and 276 genes for LUSC located on chromosome 1 (Supplementary Data 1–2 and Supplementary Fig. 3a, b). This suggests that the gain of this genomic segment impacts upon the expression of a large set of genes, including 14 genes that have been previously linked to cancer (Supplementary Fig. 3c and Supplementary Table 1). However, MCL-1 falls within the peak of frequently gained segments on 1q21 identified by GISTIC2 of the TRACERx LUAD data ($-\log10$(q value) = 4.03, Supplementary Fig. 3d, e). No significant peak on 1q21 was detected in TRACERx LUSC tumours. For this reason and due to its higher incidence[2], we focused on MCL-1 gains in LUAD.

LUAD encompasses 40% of the lung cancer patients and its 5-year overall survival across all subgroups rate averages at only 18%[18], highlighting the need of a better understanding of its molecular mechanisms in order to design new therapeutic approaches. First, we analysed MCL-1 genomic gains during tumour evolution, taking further advantage of the TRACERx study. Notably, the design of this prospective study allows the analysis of intra-tumour heterogeneity and the evolutionary trajectory of tumour genomes (Supplementary Fig. 4a). We detected clonal MCL-1 amplifications in 34/46 (74%) LUAD (e.g. CRUK034, Supplementary Fig. 4b). In 12/46 (26%) adenocarcinomas, MCL-1 gains appeared subclonally on the branches of the tumour's phylogenetic trees (e.g. CRUK001, Supplementary Fig. 4c). Given its relatively high frequency as a late event, these data suggest that the overall frequency of MCL-1 gains may be underestimated.

We then classified the LUAD samples according to TP53 status (mutant vs wild-type, Fig. 2a, b) and explored MCL-1 genomic gains in these cohorts. While the total (clonal and subclonal) MCL-1 gains were not different in these cohorts, we detected a higher frequency of subclonal events in the TP53 mutant cohort (Fig. 2c). To further analyse whether TP53 mutational status affects other genes located on chromosome 1q21, we analysed the segment size detecting no significant differences (Fig. 2d). The same analysis performed for mutant KRAS and EGFR showed no differences (Supplementary Fig. 5).

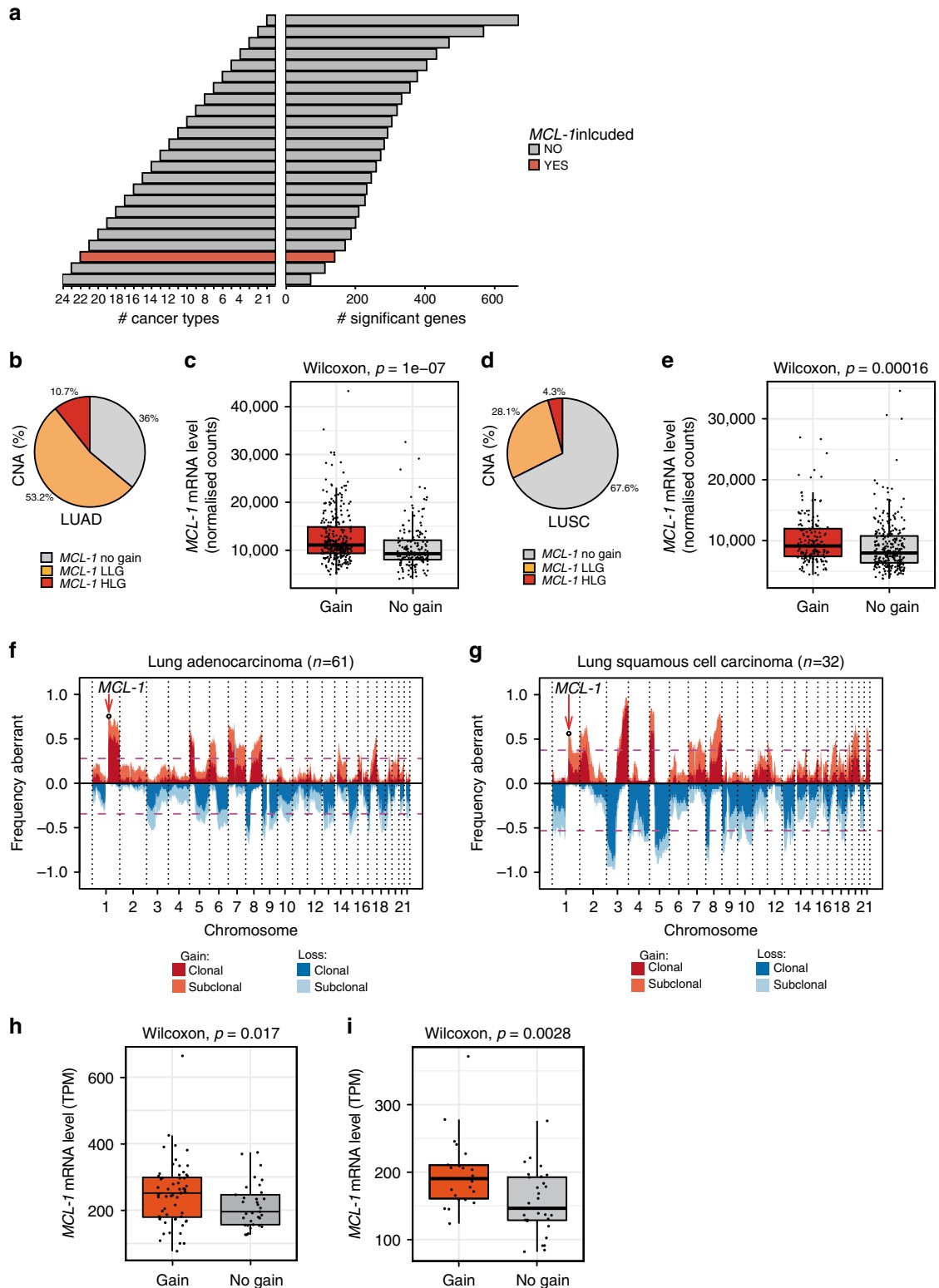

To confirm *MCL-1* genomic gains in a different cohort, we used fluorescence in situ hybridisation (FISH) with *MCL-1* probes on two tissue microarrays (TMA) of 56 and 24 primary human LUAD tissue samples. FISH analysis revealed polysomies of chromosome 1q21 and high- and low-level gain of *MCL-1* in 48.8% of the samples (Fig. 3a, b). We analysed MCL-1 protein levels in the second TMA, detecting a strong staining intensity for 10 out of 25 (40%) samples and an intermediate staining intensity

in 10/25 (40%) (Fig. 3c–e). Due to the restricted sample size no significant correlation was detected between *MCL-1* genomic gains and MCL-1 protein expression (Fig. 3f). However, none of the samples showing *MCL-1* high- or low- level gain by FISH had a weak MCL-1 protein staining intensity (Supplementary Table 2).

To understand whether MCL-1 is differentially expressed in cancer *vs* healthy tissue, we analysed MCL-1 protein levels in 11

**Fig. 1 Genomic _MCL-1_ gains in cancer. a** Number of significantly gained and expressed genes located on chromosome arm 1q in the TCGA cohort. The bars pointing to the right demonstrate the number of significant genes identified in up to 1 to 24 cancer types (right bars). The numbers are represented as cumulative counts, which means that genes significant in two cancer types were also counted in for one cancer type. _MCL-1_ was found to be significantly gained and expressed in 22 cancer types among 140 other genes. **b** _MCL-1_ gains (CNA, copy number aberrations) in TCGA LUAD samples. LLG low-level gain, HLG high-level gain. $n = 652$. **c** Correlation between _MCL-1_ CNA and mRNA levels in TCGA LUAD samples. $n = 456$ (289 gain and 167 no gain, $p = 1e-07$, $W = 17,081$). **d** _MCL-1_ gains (CNA) in TCGA LUSC samples. $n = 652$. **e** Correlation between _MCL-1_ CNA and mRNA levels in TCGA LUSC samples. $n = 441$ (164 gain and 277 no gain, $p = 1.6e-04$, $W = 18,066$). **f, g** CNA expressed as frequency in TRACERx LUAD (**f**, $n = 61$) and LUSC (**g**, $n = 32$) tumours. Shading indicates clonal status. Dashed purple lines represent threshold for frequent gains and losses. **h, i** Correlation between CNA and mRNA levels of _MCL-1_ in TRACERx LUAD (**h**) and LUSC (**i**) tumour regions. For LUAD, $n = 93$ (60 gain and 33 no gain, $p = 0.017$, $W = 1256$). For LUSC, $n = 50$ (22 gain and 28 no gain, $p = 0.0028$, $W = 448$). Data in (**c**, **e**, **h**, and **i**) are presented as box plot and were analysed by a one-sided Wilcoxon test. In the box plots, the centre line represents the median value, the limits represent the 25th and 75th percentile, the whiskers represent the minimum and maximum value of the distribution excluding outliers defined by the inter-quartile range (IQR) rule. The points represent the individual experimental values.

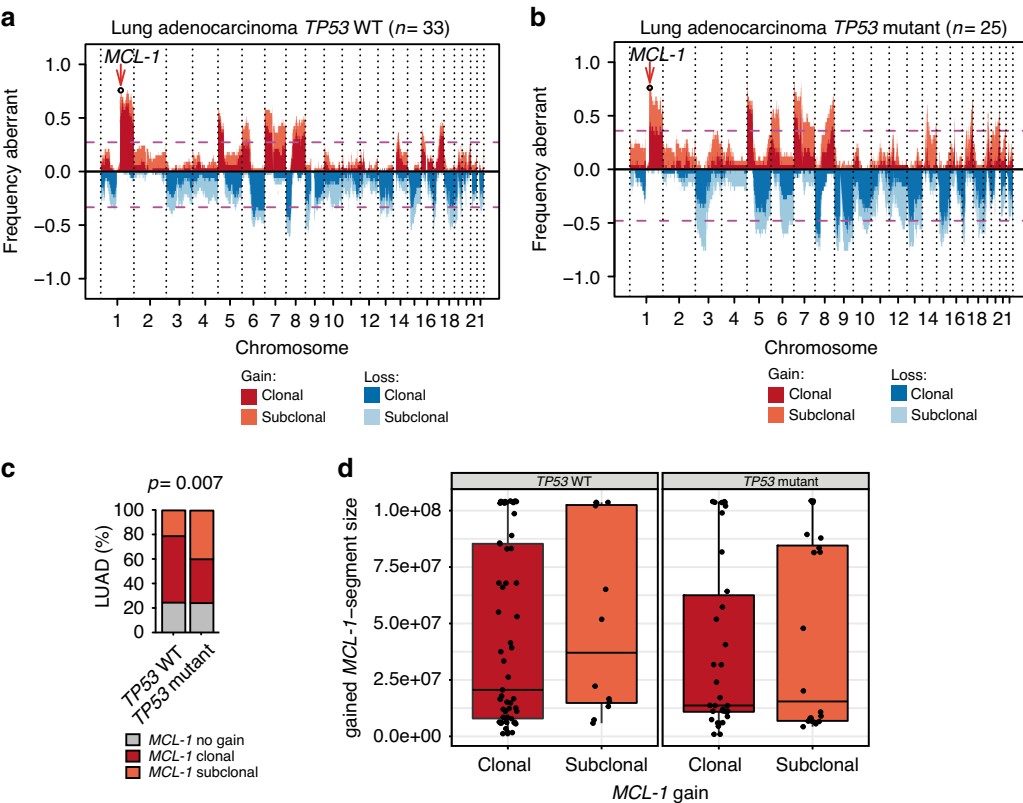

**Fig. 2 _TP53_ mutation associates with subclonal _MCL-1_ gains. a, b** CNA expressed as frequency in TRACERx LUAD grouped for _TP53_ wild-type (WT) (**a**) and mutant (**b**). Shading indicates clonal status. $n$ = number of patients. **c** Frequency distribution of _MCL-1_ gain in _TP53_ wild-type or clonal mutant TRACERx LUAD patients. Data were analysed by Pearson's chi-squared test with continuity correction ($p = 0.007$, chi-square(2) = 9.885). **d** Comparison of _MCL-1_- segment sizes between clonal and subclonal _MCL-1_ gains for _TP53_ mutant and WT samples. Two-sided Wilcoxon testing showed no significant differences. _TP53_ mut $n = 54$ (34 clonal and 20 subclonal gains, $p = 0.89$, $W = 348$) _TP53_ WT $n = 69$ (57 clonal and 12 subclonal gains, $p = 0.53$, $W = 302$). $n$ = number of samples. The data are presented as box plot, where the centre line represents the median value, the limits represent the 25th and 75th percentile, the whiskers represent the minimum and maximum value of the distribution, and the points represent the individual experimental values.

primary LUAD patient samples and matched healthy lung tissue. Notably, MCL-1 expression intensity was significantly higher in LUAD tissue than in healthy adjacent tissue (Fig. 3g, h). These results are consistent with previous reports on MCL-1 protein abundance in lung cancer[14,19,20].

Taken together, these data indicate that _MCL-1_ clonal and subclonal gains correlate with higher _MCL-1_ mRNA and MCL-1 protein levels and frequently occur in human pulmonary adenocarcinoma.

**Dissecting _MCL-1_ role in lung cancer-cell line survival.** To test whether _MCL-1_ gain represents a common feature also in cell lines, we first took advantage of online available data (COSMIC

Cell Lines Project). We detected _MCL-1_ genomic gains, which correlated with high _MCL-1_ mRNA levels in the entire cohort of LUAD cancer-cell lines (Supplementary Fig. 6).

Then, we used 11 different human NSCLC-derived cell lines (9 LUAD and 2 large cell cancer, characteristics reported in Supplementary Table 3) to test for a functional role for MCL-1 in cancer-cell survival. Genomic DNA was extracted from all the cell lines and used for sequencing and gene copy number analysis to identify genomic gains of chromosome 1q21 and specific gains of _MCL-1_. We detected a correlation between _MCL-1_ genomic gains, _MCL-1_ mRNA, and MCL-1 protein levels. This correlation was even stronger, when we focused our analysis only on LUAD cell lines (Fig. 4a and Supplementary Fig. 7).

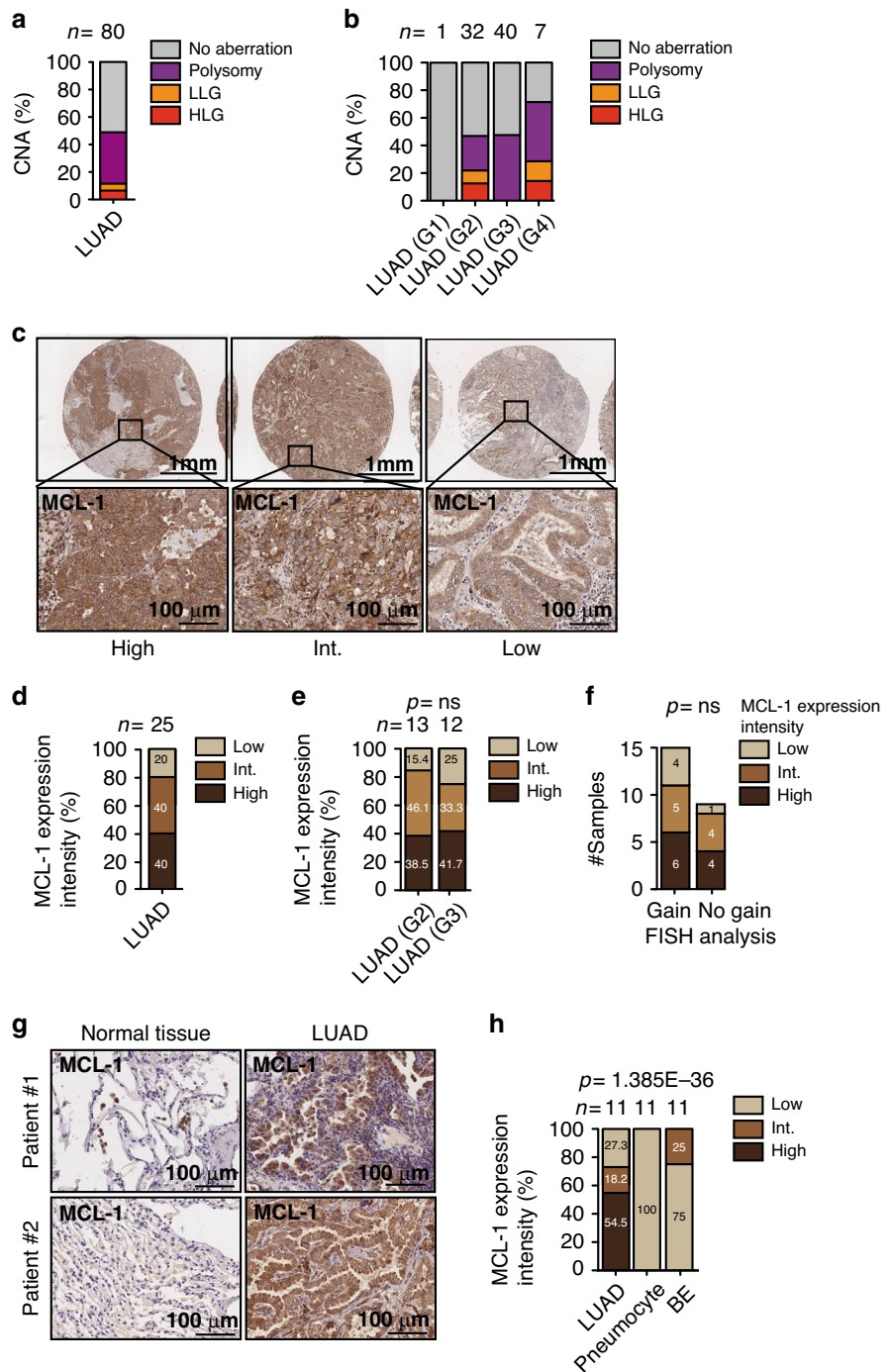

**Fig. 3 MCL-1 is over-expressed in human LUAD. a, b** Quantification of FISH analysis for *MCL-1* in LUAD TMA in all the samples (**a**) or divided by grade (**b**). *n* = 80. HLG high-level gain, LLG low level gain. **c, d** Representative images (**c**) and quantification (**d**) of MCL-1 staining in TMA of LUAD (*n* = 25). MCL-1 staining intensity was assessed and classified as low, intermediate (int.), or high. **e** Comparison of MCL-1 staining intensities between different histological grades of LUAD (G2, *n* = 13; G3, *n* = 12). Data were analysed with the Pearson's chi-squared test with continuity correction (ns, *p* = 0.0930, chi-square(2) = 4.750). **f** Association between FISH and IHC analyses on TMA samples. A two-sided Fisher's exact test was applied as statistical test (ns, *p* = 0.75). **g–h** Representative images (**g**) and quantification (**h**) of MCL-1 staining in LUAD and paired normal tissue (*n* = 11). Data were analysed by Pearson's chi-squared rest with continuity correction (*p* = 1.385E−36, chi-square(4) = 174.1). Scale bar is 1 mm or 100 µm. BE bronchial epithelium.

To examine the functional role of MCL-1 in cancer-cell survival, we treated LUAD cell lines with S63845[21], a selective MCL-1 inhibitor. Notably, the cytotoxicity of S63845 correlated with MCL-1 protein level (Fig. 4b, c).

To understand whether the other cancer-related genes identified on amplicon 1q21 (Supplementary Fig. 3c) may play a role in the sensitivity to the MCL-1 inhibitor, we analysed their expression in the cell line cohort. While we still identified a significant correlation with *MCL-1* mRNA levels, none was detected for any of the other genes analysed (Supplementary Tables 4, 5).

However, MCL-1 function also heavily depends on the balance with the alternative BCL-2 family proteins. As previously shown in other cancer cohorts[12,22], *MCL-1* gains did not elicit

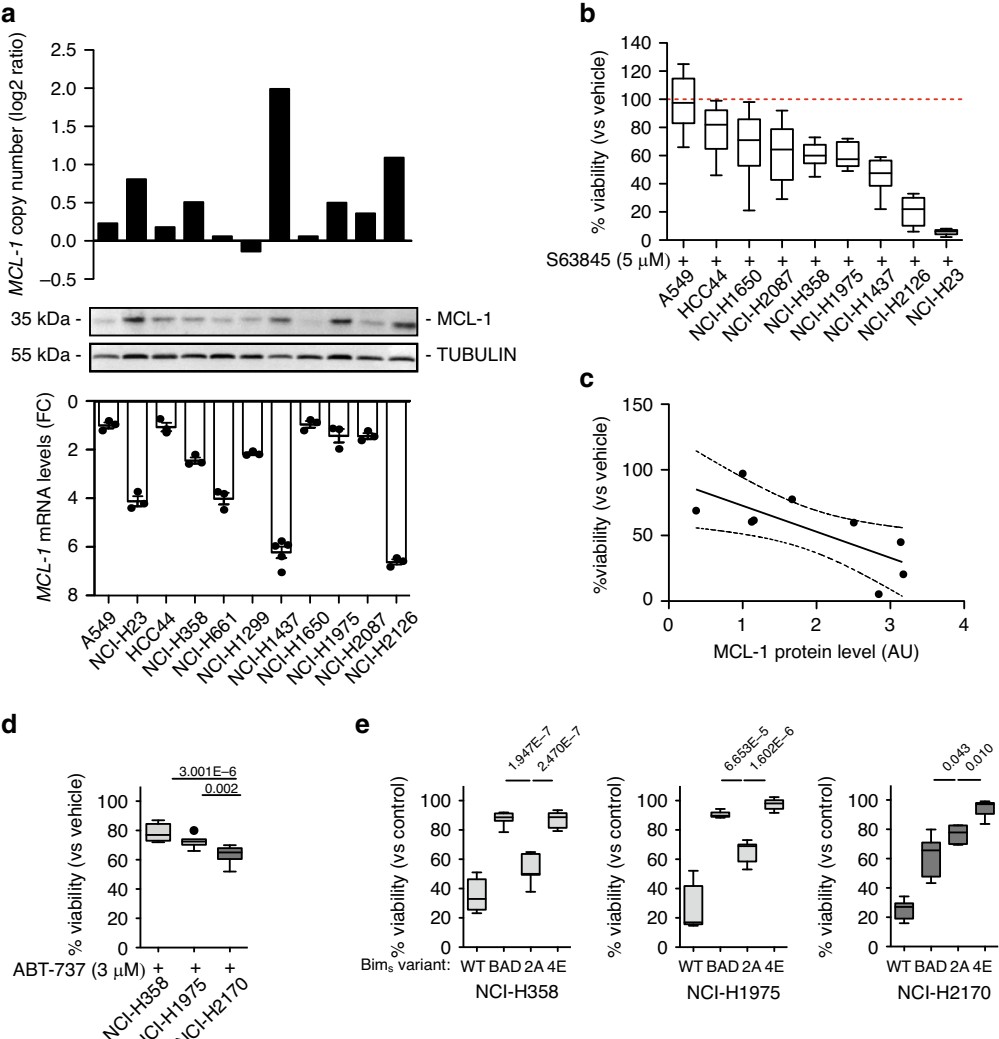

**Fig. 4 MCL-1 secures survival of lung cancer derived cell lines. a** *MCL-1* copy number variation, *MCL-1* mRNA, and MCL-1 protein levels in the cell lines used for this study. Data in the bottom panel are presented as mean ± SEM. **b** Relative viability of the indicated cell lines after 72 h treatment with 5 μM S63945, as determined by CellTiter-Glo®. Results were normalised to vehicle treated (control) cells. **c** Simple linear regression between MCL-1 protein levels (AU, arbitrary units) and cell viability after S63945 treatment (dashed lines 95% confidence interval; $R^2 = 0.5441$, $p = 0.0233$). **d** Relative viability of the cell lines after 72 h treatment with 3 μM ABT-737, as determined by CellTiter-Glo®. Results were normalised to vehicle treated (control) cells (one-way ANOVA $p = 4.540E{-}6$, $F_{(2,26)} = 20.49$). **e** Relative viability of the indicated cell lines 24 h after induction of the expression of the different $BIM_S$ variants, as determined by FACS analysis. Percentage of viable untreated cells was assigned as 100% (one-way ANOVA: NCI-H358: $p = 2.317E{-}12$, $F_{(3,24)} = 75.45$; NCI-H1975: $p = 1.354E{-}13$, $F_{(3,24)} = 97.82$; NCI-H2170: $p = 3.334E{-}12$, $F_{(3,20)} = 70.48$). In figure are reported the $p$ value from the post-hoc analysis with Bonferroni correction. Data in panel (**b**, **d**, and **e**) are presented as box plots, where the centre line represents the median value, the limits represent the 25th and 75th percentile, and the whiskers represent the minimum and maximum value of the distribution, excluding outliers defined using the IQR rule. Each cell line was assayed at least in three independent experiments.

compensatory dysregulation of other *BCL-2* family members in LUAD, irrespective of the cohort analysed (TRACERx or TCGA, Supplementary Fig. 8a, b). We used a pharmacological and a genetic approach in three representative cell lines (NCI-H358, NCI-H1975, and NCI-H2170) to dissect cancer-cell susceptibility to inhibitors of different pro-survival BCL-2 family members. Within these cell lines, we observed a heterogeneous expression pattern of pro-survival BCL-2 proteins: BCL-XL was highly expressed in all cell lines, while MCL-1 and BCL-2 protein levels were variable (Supplementary Fig. 8c). Specifically, NCI-H358 and NCI-H1975 expressed higher MCL-1 levels compared to NCI-H2170. Treatment with ABT-737, a BH3 mimetic able to inhibit BCL-2, BCL-XL, and BCL-W, but not MCL-1, was highly effective in killing the cell lines with low MCL-1 levels (Fig. 4d).

Next, we took advantage of mutant versions of the pro-apoptotic BH3-only protein BIM. We used doxycycline-inducible expression vectors for four different BIM variants, which are engineered to block: (i) MCL-1 only ($BIM_S2A$), (ii) BCL-2, BCL-XL, and BCL-W ($BIM_SBAD$), (iii) none of the pro-survival BCL-2 proteins ($BIM_S4E$, negative control), or (iv) all pro-survival BCL-2 proteins ($BIM_SWT$, positive control)[23]. We found that the MCL-1-specific version $BIM_S2A$ was significantly more potent in both of the high-MCL-1 cell lines compared to the joint BCL-2, BCL-XL, and BCL-W inhibitor ($BIM_SBAD$). Conversely, in the low-MCL-1 cell line (NCI-H2170) $BIM_SBAD$ was more effective than $BIM_S2A$ in eliciting cell death (Fig. 4e and Supplementary Fig. 8d, e). As expected, $BIM_SWT$ was the most effective in inducing apoptosis in all cell lines tested. This suggests that the survival of these lung cancer-cell lines is safeguarded by two or

more pro-survival BCL-2 proteins with MCL-1 or BCL-XL being the dominant factor. While we suggest that these effects are driven by the differences in MCL-1 expression, we cannot exclude the involvement of the different histological subtypes (LUAD for NCI-H358 and NCI-H1975 and LUSC for H2170).

Taken together, these data suggest that whenever MCL-1 is highly expressed it becomes critical for the sustained survival of lung cancer cells.

**Genetic deletion of *Mcl-1* impedes lung tumorigenesis in vivo**. We hypothesised that genetic deletion of *Mcl-1* would impair with lung tumorigenesis in vivo. To test this hypothesis, we utilised mice harbouring loxP-flanked *Mcl-1* alleles for the conditional deletion of *Mcl-1*[24]. *Mcl-1* deletion after adenoviral Cre delivery did not elicit any phenotype in the healthy lung (Supplementary Fig. 9). These mice were crossed to *lsl-Kras^{G12D/+}* mice, in which mutant *Kras* is conditionally expressed in cells after recombination of the lox-stop-lox (lsl) cassette[25]. We infected *lsl-Kras^{G12D/+} Mcl-1^{fl/fl}* and respective controls intranasally with adenovirus expressing Cre recombinase (AdCre). In AdCre infected pulmonary epithelial cells, Cre recombinase induces, at the same time, the expression of oncogenic *Kras^{G12D}* and the deletion of *Mcl-1*. We performed all the analysis 19 weeks after AdCre infection (Fig. 5a). Genetic deletion of *Mcl-1* was confirmed by RNA in situ hybridisation and genomic DNA analysis of primary cultures from tumour lesions (Fig. 5b, c and Supplementary Table 6).

To analyse the impact of *Mcl-1* deletion on tumour development, we quantified the number of lesions in all the experimental groups. Consistent with our hypothesis, we found that the deletion of *Mcl-1* caused a significant reduction in *Kras^{G12D}* driven malignant lesions compared to animals in which *Mcl-1* was not deleted (Fig. 5d, e). This was further confirmed by contrast agent-enhanced high-resolution computed tomography imaging (μ-CT) of tumour-burdened lungs (Fig. 5f, g). Notably, the changes in tumour burden were not accompanied by detectable changes in the proliferation rate across genotypes (Supplementary Fig. 10a, b). There was also no increase in the numbers of apoptotic cells in the tumours of the different genotypes (Supplementary Fig. 10c, d). This is not unexpected, as apoptotic cells would have been rapidly engulfed by phagocytic cells soon after the loss of MCL-1 protein, masking the effect of *Mcl-1* deletion at this late time point. Loss of *Mcl-1* did not affect the histological features of the proliferative lesions, such as surfactant protein (SP)-C, CC10, and mucin (Supplementary Fig. 10e–g).

To further examine the role of *Mcl-1* in lung tumorigenesis, we analysed pulmonary organoid cultures (termed pulmospheres) generated from all the experimental groups ex vivo. Of note, even loss of a single *Mcl-1* allele was sufficient to prevent pulmosphere growth over several days in culture (Fig. 5h, i).

Taken together, these data show that MCL-1 is required for the survival of lung cancer cells.

**MCL-1 inhibition delays adenocarcinoma progression in vivo**. Finally, to evaluate the potential of MCL-1 as a druggable target, we treated tumour-bearing mice with the MCL-1 inhibitor S63845[21]. To strengthen the potential clinical translation, we used the highly aggressive model of lung cancer driven by a *Kras*-mutation in combination with *p53*-deletion. In line with the human data, we detected an increase in *Mcl-1* mRNA expression in p53 deleted lesions compared to p53 WT lesions (Fig. 6a). Previous reports showed efficacy and tolerability of S36835 in the treatment of *Eμ-Myc* lymphoma[21] and triple negative (ER/PR/ERBB2 negative) breast cancer xenografts[26], the latter suggesting

a favourable pharmacokinetic profile, also in solid tumour. Notably, in vivo S36835 treatment was shown to be tolerated even in combination with chemotherapeutics[27]. *lsl-Kras^{G12D/+} p53^{fl/fl}* mice were randomly assigned to control (vehicle) and treatment groups (S63845). Starting at 10 weeks after AdCre infection, these mice were treated with 40 mg kg⁻¹ S63845 or vehicle for 5 days. At day 0 (baseline), 7, 14 and 30, tumour progression was monitored by μ-CT, examining three to five lesions per mouse (Fig. 6b). We observed a significant reduction in tumour size in the treated group compared to the vehicle control group after 14 and 30 days (Fig. 6c, d). Analysis of the lung tissue for *Mcl-1* genomic gains and *Mcl-1* mRNA levels did not detect any significant differences (Supplementary Fig. 11a–c). This is not unexpected due to the fact that S63845 treatment has been shown to increase MCL-1 protein levels (by delaying its degradation), without affecting *Mcl-1* mRNA levels[21]. This also indicates that the treatment (in this specific settings) did not actively select any specific clones (i.e. those expressing low level of *Mcl-1*). In addition, this effect was not due to differences in cell proliferation (Ki67) (Supplementary Fig. 11d, e). There was also no marked increase in cleaved caspase 3 staining, presumably because tumour cells undergoing apoptosis are rapidly phagocytosed (Supplementary Fig. 11f, g). Taken together, these data suggest that MCL-1 represents a druggable target to treat LUAD.

## Discussion

We show that genomic gains of the pro-survival *MCL-1* gene occur with high frequency during tumour evolution in pulmonary adenocarcinoma. Using murine tumour models, the data provided here demonstrate the suitability of MCL-1 as a rational drug target for the treatment of pulmonary adenocarcinoma.

The chromosome locus 1q21, which harbours the *MCL-1* gene, shows frequent copy number gains in lung cancer[17]. We confirmed these data in three independent cohorts (publicly available datasets TCGA and TRACERx, and an in-house tissue microarray). However, gain in *MCL-1* co-occurred with many other genes located on amplicon 1q21. While we do not exclude the involvement of these other genes in cancer initiation and/or progression, we hypothesised that the resistance to apoptosis mediated by MCL-1 serves as a critical oncogene addiction in lung cancer and possibly other cancer entities[12,22]. We argue that as an addiction, tumours favour the presence of MCL-1 and, when present, they become highly dependent on it.

Gains of *MCL-1* were predominantly clonal, occurring on the trunk of each tumour's phylogenetic tree. Interestingly, we observed higher frequencies of *MCL-1* subclonal gains in lung tumours following clonal *TP53* mutation. This could therefore simply reflect increased chromosomal instability. However, conceivably, this could also indicate that higher levels of MCL-1 are selected for by the tumour cells in response to mounting oncogenic stress that occurs when *TP53* is mutated. This notion is supported by our in vivo data, showing that *Mcl-1* levels are higher in established *p53* deleted tumours (compared to *p53* WT tumours) and MCL-1 inhibition delays tumour progression. Taken together, these data indicate that fully established tumour lesions lacking functional p53 require, at least in part, MCL-1 for sustained survival.

In line with previous reports in alternative cancer cohorts[12,22], genomic gains of *MCL-1* occurred at higher frequency than any other *BCL-2* family member and did not cause compensatory deregulation of other BCL-2 protein family members, as detected in the TCGA and TRACERx datasets. This has already been shown in hepatocellular carcinoma[28], B cell-non-Hodgkin's lymphoma[29] and T cell lymphoma[23,30,31]. However, dependencies on other pro-survival BCL-2 family members may exist

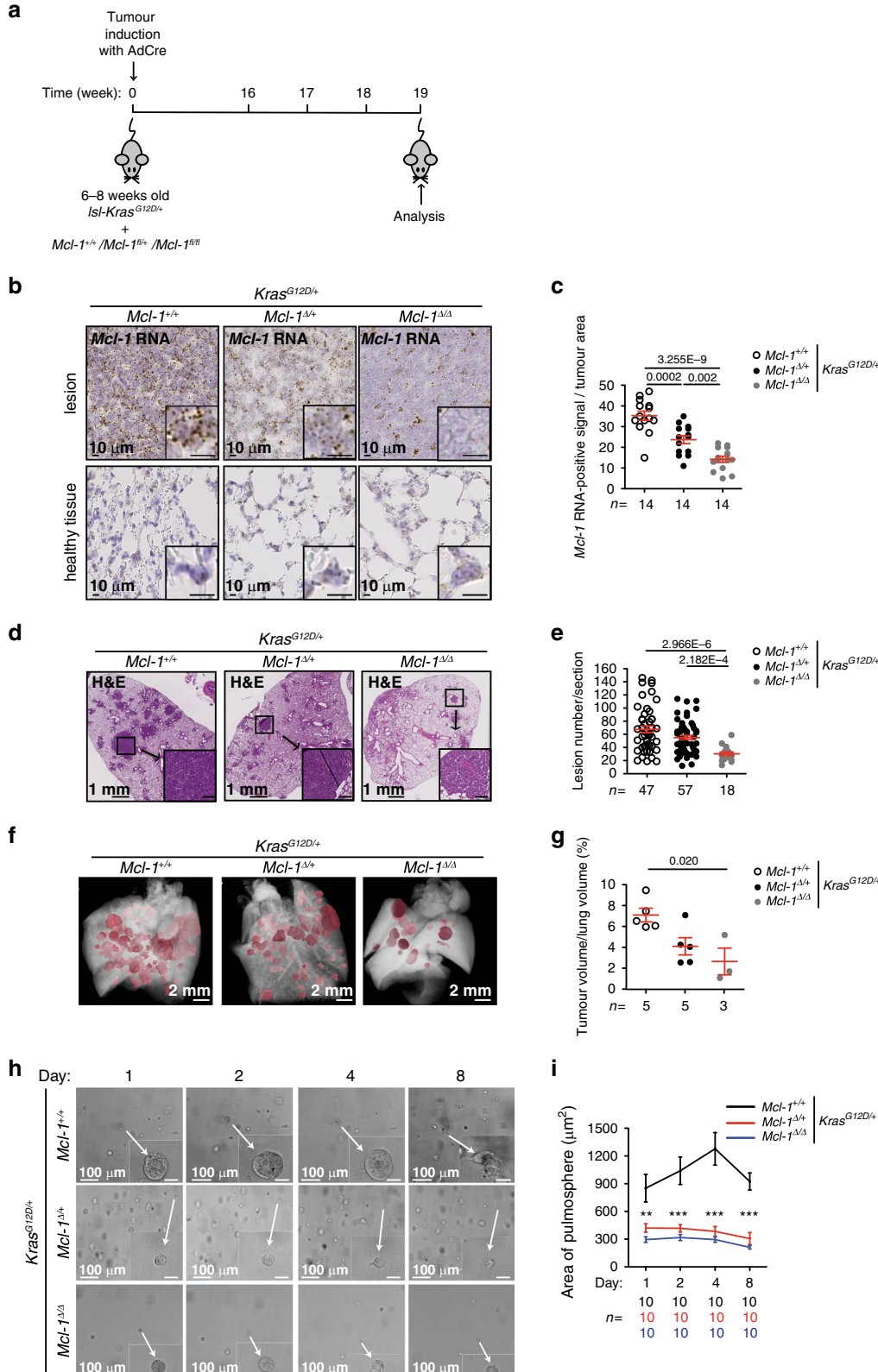

depending on the type of cancer and treatment used, as recently reported data on breast cancer showed a positive correlation between not only *MCL-1* but also *BCL-2A1* expression and an inverse association with *BCL-2* expression[26]. Our in vitro data showed that inhibition of BCL-XL, BCL-W and BCL-2 was less

effective than MCL-1 inhibition in killing cell lines expressing high levels of MCL-1, further supporting the addiction of lung tumours to MCL-1.

Notably, gains of *MCL-1* were associated with a significant increase in its mRNA and protein expression (in primary samples

**Fig. 5 Pro-survival *Mcl-1* is critical for LUAD development. a** Experimental design. **b**, **c** Representative images (**b**) and quantification (**c**) of *Mcl-1* mRNA in situ hybridisation in lesions and adjacent healthy tissue. Inserts are higher magnification of corresponding images. $n$ = number of lesions. Data were analysed by one-way ANOVA ($p = 6.600E{-}9$, $F_{(2,39)} = 31.73$, $p$ values from post-hoc analysis with Bonferroni correction are reported in figure).
**d** Representative H&E staining of lung tissue sections at 19 weeks after AdCre virus infection. Inserts are higher magnification of indicated areas. Scale bar represents 2 mm, 1 mm, 100 μm, or 10 μm. **e** Number of lesions per section. $n$ = number of sections. Three sections per animal, separated by 100 μm, were assessed. Data were analysed by Kruskall–Wallis test ($p = 5.544E{-}6$, $p$ values from post-hoc analysis with Dunn's correction are reported in figure).
**f** Representative renderings of the μ-CT analysis. **g** Quantification of tumour volume expressed as percentage of the lung volume. $n$ = number of mice. Data were analysed by one-way ANOVA ($p = 0.0145$, $F_{(2,10)} = 6.656$, $p$ values from post-hoc analysis with Bonferroni correction are reported in figure). Data in panel (**c**, **e**, and **g**) are representative of four independent experiments and are presented as dot plot and show mean ± SEM. **h** Representative images of pulmspheres generated from dissected tumour tissue (2 lesions from 2 mice/genotype). Inserts showing individual pulmspheres. Scale bar is 25 μm or 100 μm. **i** Quantification of pulmosphere area (μm$^2$) over the cultivation period. $n$ = number of pulmspheres. Data are presented as mean ± SEM and were analysed by repeated measure two-way ANOVA (for time, genotype, and interaction effects, $p < 0.0001$). $p$ values from post-hoc analysis with Bonferroni correction are reported in the panel (**p < 0.01 and ***p < 0.001).

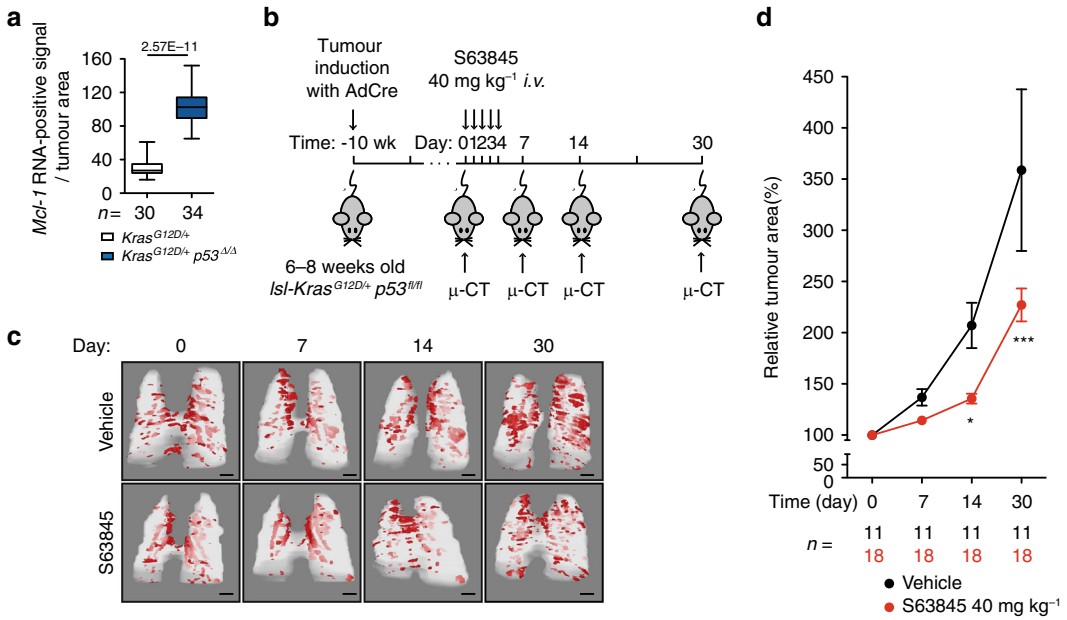

**Fig. 6 MCL-1 inhibitor delays progression of LUAD in vivo. a** Quantification of *Mcl-1* mRNA in situ hybridisation in lesions from *lsl-Kras$^{G12D/+}$* and *lsl-Kras$^{G12D/+}$;p53$^{fl/fl}$* mice. $n$ = number of lesions. Data are presented as box plot and were analysed by two-sided Mann-Whitney test ($p = 2.57E{-}11$). In the box plot, the centre line represents the median value, the limits represent the 25th and 75th percentile, and the whiskers represent the minimum and maximum value of the distribution. **b** Experimental design. Wk- week. **c** Representative rendering of μ-CT analysis. Scale bars represent 2 mm.
**d** Quantification of tumour growth is expressed as percentage of the tumour volume on day 0. $n$ = individual tumours. Three to five tumours per mouse were assessed. Data are presented as mean ± SEM and were analysed by repeated measure two-way ANOVA (time effect $p < 0.0001$, treatment effect $p = 0.001$, interaction effect $p = 0.0006$). Post-hoc analysis with Bonferroni correction showed a significant effect for treatment at day 14 (*$p < 0.05$) and day 30 (***$p < 0.001$).

and cell lines), consistent with functional consequences. Indeed, in vitro data showed that cell line susceptibility to MCL-1 inhibition significantly correlated with MCL-1 expression levels. While recent work showed that solid cancers respond only moderately to MCL-1 inhibitor S36835 single treatment[21], we suggest that targeting MCL-1 might be an effective approach on in its own[15,20]. This is in line with previous reports of successful treatment with S36835 in *Eμ-Myc* lymphoma[21] or triple negative (ER/PR/ERBB2 negative) breast cancer xenografts[26]. Alternatively, MCL-1 inhibition might be used to lower the apoptotic threshold in combination with other treatment strategies. In line with this notion, Nangia et al. showed that simultaneous inhibition of MCL-1 and MAPK provides better response rates in NSCLC. This approach is based on a MAPK-dependent increase in pro-apoptotic BIM, which can inhibit the non-targeted pro-survival BCL-2 family member(s) present in the tumour cells[32]. Our data provide evidence that LUAD exhibit elevated MCL-1 levels by genetic amplification *per se* and not as a result of treatment.

In summary, our data show that genomic gains of *MCL-1* during lung cancer evolution represent a frequent and functionally important survival mechanism for pulmonary adenocarcinoma that represents a rational drug target for the treatment of NSCLC.

## Methods

**Clonal evolution of *MCL-1* during lung cancer progression.** To determine the clonal status of *MCL-1* gains during tumour evolution, multi-region copy number data from the TRACERx study[16] was used. The TRACERx study was conducted under the favourable opinion from the NRES Committee London—Camden & Islington Research Ethics Committee[16]. The copy number values were adjusted by ploidy and log-transformed. This leads to the identification of gains as genomic segments exhibiting a copy number state > log2(2.5/2). Gains with a copy number state > log2(4/2) were classified as high-level gains. Clonal gains were ubiquitously identified across every tumour region sequenced from a given tumour, while subclonal gains were identified as present in at least one tumour region, but not all. Based on presence or absence within individual regions, gains were mapped to previously described phylogenetic trees[16]. To identify significant frequent gains and losses across the cohort, the 95% quantile of the corresponding null distribution

was used as a threshold. The null distributions for gains and losses respectively were calculated under the assumption that gains and losses occur randomly along the genome. A threshold of 20 Mb was used to distinguish between focal and broad gains. In order to check which of the genes located in the gained area around *MCL-1* show an increase in expression, the RNAseq data from the TRACERx cohort were analysed. The mRNA levels were measured as TPM (Transcripts Per Kilobase Million) values. A one-sided Wilcoxon test was applied to test for a significant difference between expression levels of samples with gains versus no gains. A combined list of previously reported lung cancer genes[16] was used to find putative cancer genes in the significant differentially expressed genes.

**Mice**. The previously established *lox-STOP-lox-Kras^{G12D}* (*lsl-Kras^{G12D}*)[25], *Mcl-1^{floxed}*[24], and *p53^{floxed}*[33] mice were used to generate *lsl-Kras^{G12D};Mcl-1^{floxed}* mice and *lsl-Kras^{G12D};p53^{floxed}* mice. Mice were maintained under pathogen-free conditions and assigned to experimental groups based on their genotype after birth. There were no other factors that determined group selection. All mice were housed up to five per cage at 20–22 °C, 45–60% humidity, kept on a 12-h light/dark cycle and were given *ad libitum* access to food and water. We randomly assigned mice to a specific group depending only on the genotype. No mice were excluded from the statistical analyses. All animal experiments were conducted in compliance with protocols approved by the District Government of Upper Bavaria in line with the animal ethics committee guidelines (Az: 55.2-1-54-2532- 55-12 and ROB-55.2-2532.Vet_02-14-69).

**TCGA data analysis**. The genomic data of 24 different cancer types (BLCA, BRCA, CESC, CHOL, DLBC, ESCA, GBM, HNSC, LGG, LIHC, LUAD, LUSC, MESO, OV, PAAD, PCPG, PRAD, READ, SARC, STAD, TGCT, THCA, THYM, UCE, UCS) of The Cancer Genome Atlas (TCGA) was analysed in order to understand the preponderance of genomic gains of *MCL-1* in cancer. Copy number gains and losses were classified in the same way as in the TRACERx dataset. Normalised mRNA counts were used in addition to the copy number classification to identify genes that are significantly gained and expressed on chromosome arm 1q. The resulting genes were then compared between cancer types to identify how commonly *MCL-1* is significantly gained and expressed across cancer types.

The TCGA data of the NSCLC subtypes LUAD and LUSC has been further analysed to identify the frequency of high- and low-level gains of *MCL-1*, respectively. A one-sided Wilcoxon test was applied to test for a significant difference between *MCL-1* expression levels of samples with gains versus no gains.

To build the heatmap for BCL-2 family, the expression data from TGCA-LUAD were medium centralized (both row and column, range −3 to 3) and then complete-linkage clustered before being plotted into the heatmap. The unit of expression is "log2(count + 1)" and the count represents RSEM[34] normalised count. Accessed on 20.10.2017.

**TMA-construction**. Formalin-fixed paraffin-embedded (FFPE) tumour samples were assembled into a tissue microarray (TMA) using a Tissue Microarrayer (Alpha Metrix Biotech, Rödermark, Germany), extracting tumour samples with a core size of 1 mm per tumour block and case. All samples of a respective tumour region were extracted from areas exhibiting a high tumour/stroma ratio. Obvious inflammatory hotspots (such as lymph follicles or areas of ulceration) were avoided. The first investigated TMA was provided by Lungbiobank Heidelberg, member of the Biomaterialbank Heidelberg (BMBH) and the biobank platform of the German Centre for Lung Research (DZL). The TMA contained 102 tumour samples from 56 pulmonary adenocarcinomas, resected between 2005 and 2008 at the Thoraxklinik Heidelberg, Germany and this study had been approved by the local human ethics committee (no. 206/2005). The second study investigated TMA contained 75 tumour samples from 25 pulmonary adenocarcinomas. For this study, materials from the archive of the Institute of Pathology of the TUM was used, anonymised and provided by the MRI/TUM-biobank (approved by ethical committee Nr. 553/15 S). All patients had given written consent to the use of surgically removed tissue for scientific purposes.

**MCL-1 fluorescence in situ hybridisation**. *MCL-1* gene copy number status was investigated using the ZytoLight SPEC *MCL-1/1p12* Dual Colour Probe (Zytovision, Bremerhaven, Germany), which used according to the manufacturer's instructions. The probe contained green-labelled polynucleotides (ZyGreen: excitation at 503 nm and emission at 528 nm, similar to FITC), which target the human *MCL-1* gene in 1q21.3, and orange-labelled polynucleotides (ZyOrange: excitation at 547 nm and emission at 572 nm, similar to rhodamine), which target sequences in the chromosomal region 1p12. *MCL-1* gene copy number status was assessed. *MCL-1* gene gain was defined as MCL-1/1p12 ratio ≥2.0, with high-level gain defined as ≥4.0 and low level gain defined as ≥2.0 and ≤4.0. Polysomy was defined as average *MCL-1* gene copy number >3 signals/cell. At least 20 neoplastic cells per core were evaluated. Areas with overlapping tumour cells were excluded from the analysis.

**MCL-1 immunohistochemistry and evaluation**. Primary LUAD tissues were stained with antibodies against MCL-1 (1:400, Proteintech,16225-1-AP) using an automated staining system (Bond Max, Leica microsystems). Expression was quantified with respect to the number of positive tumour cells. Staining intensity

was assessed in a semi-quantitative manner by experimenters blinded to the diagnosis taking alveolar macrophages as reference for strong staining, as they showed a diffuse and distinct MCL-1 positivity throughout all cases. Representative images of strong, intermediate, and low intensity staining of MCL-1 are reported in Fig. 3c.

**In vitro experiments**. Human NSCLC cell lines: A549, NCI-H23, NCI-H358, NCI-H1437, NCI-H1650, NCI-H1975, NCI-H2087, NCI-H2126, NCI-H661, NCI-H1299, and NCI-H2170 were obtained from ATCC and HCC44 was purchased from Leibniz Institute DSMZ. The cell lines were authenticated with STR profiling analysis by ATCC. HCC44 was analysed with STR profiling and verified by DSMZ profile database[35]. The cells were cultured at 37 °C in complete media in an atmosphere containing 5% $CO_2$ (RPMI medium 1640 plus 10% foetal bovine serum (FBS), 2 mM L-Glutamine, 10 mM HEPES, 1 mM sodium pyruvate, 0.15% sodium bicarbonate, 25 mM glucose, and 1% penicillin/streptomycin).

Genomic DNA from the cell lines was extracted with the Blood & Cell Culture DNA Kit kit (Qiagen) according to manufacturer's guidelines. Genomic DNA from peripheral mononuclear blood cells of a healthy individual was used as control. Library preparation was performed with an average of 200 ng DNA per sample using the NEBNext Ultra II FS DNA Library Prep Kit for Illumina. Samples were sequenced on an Illumina NextSeq system, resulting in ~20 Mio. single-end, 59 bp long, reads per sample. Resulting sequencing reads were trimmed using Trimmomatic and mapped to the human reference genome GRCh38.p12 using bwa mem. The GATK toolkit was used for base recalibration. Copy number alterations were called by HMMCopy. Total RNA isolation from the cell lines was performed using the Direct-zol MiniPrep Plus kit (Zymo Research) according to manufacturer's instructions. cDNA was synthetised with QuantiTect Reverse Transcription Kit (Qiagen) and was analysed by qRT-PCR using the GoTaq qPCR Master Mix (Promega) on the LightCycler® 480 Instrument (Roche Life Science). The following primers were used: actin (for: AGAAAATCTGGCACCACACC, rev: AGAGGCGTACAGGGATAGCA), *ABL2* (for: CACAGAGACCGGCTTCAATA, rev: TCATTTAGTGCCTGGGGTTC), *ARNT* (for: AGGGCTGGATTTTGATGATG, rev: CGCCGTTCAATTTCACTGT), *BCL9* (for: GAAGACTCCAGCCAAAGTGG, rev: GTGTGTTCAGAGGCGCTGT), *CDC73* (for: AATCTGTAACGGAGGGTGCAT, rev: AAGAGGTGGTAGCTGCAGGA), *MCL-1* (for: AGAAAGCTGCATCGAACCAT, rev: CCAGCTCCTACTCCAGCAAC), *MLLT11* (for: AGGCCTGGGTCTGTCAGATA, rev: CAATGGGAGCTCTCCAGAAG), *PDE4DIP* (for: CCACACCCTGGATGAGAGAT, rev: GGAGGGCCTCGATCTTTAGT), *PRCC* (for: AGGAAAGAGCCCGTGAAGAT, rev: CAGGTTTTTAGGTTGGGGAAG), *RIT-1* (for: AGTTCATCAGCCACCGATTC, rev: CATATACTGGTCCCGCATGG), *SETDB1* (for: CCGGCCTACAGAAATAATTGA, rev: CCTGGGAACTGCTCTTCTTG), *SDHC* (for: TCAAACCGTCCTCTGTCTCC, rev: AGAGACCCCTGCACTCAAAG), *TPM3* (for: AAACTCAAGGAGGCAGAGACC, rev: AGGTCAAGCAGGGTCTGGT), *TPR* (for: TGAGCAGCTTGAGAAACTTCA, rev: TGCTTTTGAGTTGTTGGCAGT).

Total protein extracts were prepared using RIPA buffer (Cell Signaling cat# 9806) containing protease inhibitors (complete protease inhibitor cocktail, Roche) and protein concentrations were determined by using the BCA assay (Thermo Scientific cat# 23225). Antibodies used were: anti-MCL-1[36] (1:1000, 19C4-15, rat monoclonal, gift from David Huang, WEHI), anti-BCL-2 (1:1000, Bcl-2-100, produced at WEHI), BCL-XL (1:1000, BD Bioscience, cat# 610212), BIM (1:1000, Enzo Life Science, cat# ADI-AAP-33-E), Caspase-3 (1:1000, Cell Signaling, cat# 9662), and KRAS (1:1000, SigmaAldrich, cat #MABS194, clone# 234-4.2). β-actin (1:40000, Cell Signaling, cat #4970, clone #13E5) and α-tubulin (1:5000, Cell Signaling, cat#9099) were used as loading controls. Uncropped gel images are provided as Supplementary Figs. 12–15.

For the pharmacological treatment, the cell lines were seeded into 96-well plates. They were exposed for 72 h to 3 μM ABT-737 (Active Biochem, A-1002) or 5 μM S63845 (Active Biochem, A-6044). Cell viability was determined by using the CellTiter-Glo® assay (Promega, cat# G7175).

The doxycycline-responsive lentiviral vector pFTRE3G_pGK3G_GFP encoding different BIM_s variants have been previously described[23]. The lentivirally transduced NSCLC cells were seeded and on the next day treated with 1 μg/mL doxycycline to induce the expression of the BIM_S variants. After 24 and 48 h, cells were collected in FACS buffer (PBS + 3% FBS) and the percentage of viable cells was determined by propidium iodide (PI) staining (3 μg/mL) and FACS analysis. PI-negative and green fluorescent protein (GFP)-positive cells were considered to be live cells (gating strategy is shown in Supplementary Fig. 16).

**In vivo experiments**. At 6–8 weeks of age, mice were infected with $5 \times 10^6$ plaque forming units (PFU) of AdCre[37] in two 62.5 μL intranasal instillations[25]. For the experiments reported in Fig. 5 and Supplementary Figs. 9–10, mice of the different genotypes (*Kras^{G12D/+};Mcl-1^{+/+}*, *Kras^{G12D/+};Mcl-1^{Δ/+}*, and *Kras^{G12D/+};Mcl-1^{Δ/Δ}*) were analysed 19 weeks after infection. Ethics approval number: 55.2-1-54-2532-55-12. For the experiments reported in Fig. 6 and Supplementary Fig. 11, *Kras^{G12D/+}p53^{Δ/Δ}* mice were randomised into two different groups at 10 weeks after infection: treatment with S63845 (40 mg kg⁻¹ body weight per day for 5 consecutive days) and vehicle. S63845 (from Active Biochem, A-6044) was formulated in 25 mM HCl, and 20% 2-hydroxy propyl β-cyclodextrin (SigmaAldrich) and administered intravenously (*i.v.*)[21]. Ethics approval number: ROB-55.2-2532.Vet_02-14-69.

***Mcl-1* analysis in mouse lungs**. In order to check *Mcl-1* mRNA expression in the lesions, RNA in situ hybridisation based RNAscope® 2.0 FFPE assay was used with the target probe (*mmMcl-1*, cat# 317241) according to the manual (Advanced Cell Diagnostics, Inc, Rev 20120921)[38]. *mmUBC*, cat# 310771 and *dapB*, cat# 310043 served as technical positive and negative controls, respectively. Quantification of ISH was carried out manually by experimenters blinded to the genotype. The positive signal for *Mcl-1* mRNA was measured in individual 2000 $\mu m^2$ lesion areas (approximately 20 cells each).

In order to evaluate *Mcl-1* copy number in lesion from $Kras^{G12D/+}p53^{\Delta/\Delta}$ deletion, we extracted genomic DNA from lesion from FFPE slides with the prepGEM Universal (MicroGEM International) following the manufacturer´s instructions. Extracts were diluted to 5 ng/µL and analysed by Taqman® assay, as described below.

**Tumour burden analysis**. All lungs were sectioned into 3 step-sections at 100 µm intervals. The third sections of each step-sections were H&E stained and used for tumour burden analysis. The sections were scanned with a SCN400 slide scanner (Leica microsystems) and the area ($\mu m^2$) of individual proliferative lesions and total lung area ($\mu m^2$) on the sections were determined by manually defining the lesion within a lung using the free-hand mode of the Tissue IA image analysis software and/or Aperio imagescope (Leica microsystems). The total numbers of the lesions on each section were also counted. Three H&E stained sections per animal were analysed.

**Ex vivo micro-computed tomography (µ-CT)**. The commercial lab-based system Zeiss Xradia 500 Versa by Carl Zeiss (Carl Zeiss Microscopy) was used for X-ray µ-CT analysis. The system was driven with a tube voltage of 40 kV and a current of 63 µA. The detector assembly includes a switchable-objective lens unit (×0.39, ×0.4, ×20) coupled to a charge coupled device (CCD) camera made of 2048 × 2048 pixels for a pixel size of 13.5 µm. With the 0.39x-demagnifying objective and adequate settings of the distances from the sample to source and detector, instantaneously, the samples of 1–1.5 cm in size were entirely recorded across the field of view (FOV) granted by the detector. Accordingly, the scans were completed with effective pixel sizes in a range of 12–13 µm by recording 1201 projections over 360 degrees. The reconstruction of the tomograms was set to occur automatically with the dedicated software and performed properly after each measurement. The tumour burden was assessed by experimenters blinded to the genotype for segmentation owing to the substantial contrast expressed between the lung healthy lung tissue and the lung lesions. For the purpose of quantitative analysis, the segmentation was performed with VGStudioMax² 2.1 (Volume Graphics GmbH, Heidelberg, Germany).

**Evaluation of proliferative lesion histology**. One slide per mouse, in total 8–11 mice per genotype, were analysed histologically and classified as hyperplasia, adenoma or adenocarcinoma[25,39]. Specifically, hyperplasia is defined by increased cellularity without atypia, enlarged cells, and normal architecture of bronchioles and alveoli; adenoma is characterised by well circumscribed areas consisting of cuboidal to columnar cells lining alveoli, uniform population of epithelial cells with round nuclei, size less than 5 mm in diameter, and absence of vascular invasion; adenocarcinoma is defined by nuclear enlargement and prominent nucleoli, increased mitotic rate, size over 5 mm in diameter, and invasion of vessels. If not otherwise indicated, proliferative lesions included hyperplasia, adenoma, and adenocarcinoma.

**Immunohistochemistry**. Immunohistochemical staining was performed using an automated staining system, Bond Max (Leica microsystems). Antibodies against the following target proteins were used: Ki67 (1:200, ThermoFisher, cat #RM-9106, clone #SP6), cleaved caspase-3 (Asp175) (1:300, Cell Signaling, cat# 9661), CC10 (T18) (1:100, Santa-Cruz, cat #sc-9772), SP-C (C19) (1:500, Santa-Cruz, cat #sc-7705). For quantification of staining and image acquisition, slides were scanned using a SCN400 slide scanner (Leica microsystems) and analysed using Tissue IA image analysis software (Slidepath, Leica microsystems) and/or Imagescope Aperio (Leica micropsystems) by experimenters blinded to the genotype.

**Primary culture and 3D lung organoids (pulmosphere) culture**. Single cells were generated from freshly excised lung tumours. The primary cells were cultured in complete PneumaCult™-Ex medium (STEMCELL technology, cat# 05008) at 37 ° C in an atmosphere containing 5% $CO_2$.

Genomic DNA was extracted from these primary cultures with the prepGEM Universal (MicroGEM) following the manufacturer's instructions. To evaluate *Mcl-1* copy number, we used Taqman® copy number assays (ThermoFischer) with the *Mcl-1* probe (assay ID: Mm00544723_cn) and the reference probe *Tfrc* (cat#: 4458366). Data were analysed with CopyCaller® software (Applied Biosystems).

1000 live cells were re-suspended in 100 µL of PneumaCult™-Ex Medium on ice. The cell-matrigel mixture was then plated in µ-Plate 96-well ibiTreat (ibidi) and incubated at 37 °C for half an hour. The semi-solid culture was topped with another 200 µL of PneumaCult™-Ex medium and incubated for 3 days. The pulmospheres were observed in Leica microscope for 8 days, the position of pulmosphere was fixed with defined x and y axis and then the growth was monitored on day 1, 2, 4 and 8.

**In vivo µ-CT of S63845/vehicle treated mice**. Mice were anesthetised with 2% isoflurane as maintenance dose after induction with 4% isoflurane. µ-CT images of free-breathing animals were acquired using an in vivo µ-CT scanner (Skyscan 1176, Bruker, Kontich, Belgium). Animals were scanned at 50 kV X-ray source voltage and 180 µA source current combined with 0.5 mm aluminium filter, acquiring 7 projections/position with 0.7° increments over 180°. This resulted in a scanning time of approximately 8 min with 35 µm image resolution. Image-based data sorting using DataViewer software (Version 1.5.6.2, Bruker, Kontich, Belgium) was employed yielding four 3D reconstructed datasets corresponding to four different phases of the breathing cycle. The breathing cycles were determined using a coupled camera using thorax movements. Subsequently, µ-CT data were retrospectively reconstructed using NRecon (Version 1.7.3.2, Bruker, Kontich, Belgium), DataViewer and CTAnalyser (Version 1.17.7.2, Bruker, Kontich, Belgium) for the generation of 3D model representing the tumour burden in the lung. Ethics approval number: ROB-55.2-2532.Vet_02-14-69. The reconstructed images were viewed and measured with Horos software by experimenters blinded to the treatment. The area ($mm^2$) of individual tumour was determined by manually defining the biggest tumour area using pencil tool of the software. Three to five tumour lesions that were visible at every time point were selected per mouse. The baseline measurement of individual tumour was assigned as 100%.

**Statistical analysis and data presentation**. The data are presented as dot plot or box plot. For the dot plots, mean and SEM are reported in figure. In the box plots, the centre line represents the median value, the limits represent the 25th and 75th percentile, and the whiskers extend from the box to the largest and lowest value no further than 1.5 * IQR away from the box, where IQR is the inter-quartile range.

The data analysis was performed in the R statistical environment version > = 3.3.1 and with GraphPad Prism software. Comparisons between two groups were made using one-sided Wilcoxon test, whereas for more than two groups one-way ANOVA followed by post-hoc analysis with Bonferroni correction was used unless differently stated. A permutation method was applied to calculate the null distributions for the frequency of gains and losses followed by 95% quantile calculation to identify frequently gained and lost events. A list of published cancer driver genes[16] was used to check which of the identified differentially expressed genes were reported previously. For analyses of distribution, the chi-squared test was used. For analysis of the effects of genotype (pulmosphere) or treatment (S63845) across time, repeated measure two-way ANOVA with Bonferroni post-hoc analysis was used. All values are presented as means ± SEM if not otherwise indicated. For all statistical tests statistical significance was determined if the *p* value was less than 0.05, unless otherwise stated.

**Reporting summary**. Further information on research design is available in the Nature Research Reporting Summary linked to this article.

## Data availability

The data that support the findings of this study are available from the corresponding author upon reasonable request. The data generated by The Cancer Genome Atlas pilot project established by the NCI and the National Human Genome Research Institute was downloaded. The data were retrieved through database of Genotypes and Phenotypes (dbGaP) authorisation (accession no. phs000854.v3.p8). Information about TCGA and the investigators and institutions who constitute the TCGA research network can be found at https://cancergenome.nih.gov/. The genomic TRACERx data can be downloaded from the European Genome-Phenome Archive (EGA), which is hosted by the European Bioinformatics Institute (EBI) and the Centre for Genomic Regulation (CRG), under the accession number EGAS00001002247. The data have been processed as described in ref. [16]. The remaining data are available within the Article, Supplementary Information or available from the authors upon request.

## Code availability

The data collection and processing were performed in the R statistical environment version >= 3.3.1. Code for the data analysis can be downloaded from https://github.com/McGranahanLab/MCL1amplification.

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

## Acknowledgements

The authors thank the members of the TRACERx consortium for participating in this study. The results published here are in part based upon data generated by The Cancer Genome Atlas pilot project established by the NCI and the National Human Genome Research Institute. The authors would like to thank S Schwamberger and I Konrad for mouse handling, and Life Science Editors for editing assistance. We thank David Huang for gift of antibodies. The authors would like to thank Gewebebank des Klinikums rechts der Isar und der Technischen Universität München, A Terron Kwiatkowski, A Jacob, R. Öllinger, and G Sannino for the excellent technical support. We thank Martin Schuler for critical discussion of the data. Grant Support Grant support by the Mildred Scheel Professorship program from the Deutsche Krebshilfe (program #70113247), the Deutsche Forschungsgemeinschaft (DFG, German Research Foundation – FOR 2036, SFB 1335 (Project-ID 360372040), SFB 1371), German Consortium for Translational Cancer Research (DKTK) to P.J.J.; the Deutsche Forschungsgemeinschaft (DFG, German Research Foundation - SFB 1335 (Project-ID 360372040), SFB 1371) to K.S. A.S. is supported by National Health and Medical Research Council (NHMRC) project grant 1143105, NHMRC program grant 1016701, NHMRC fellowship 1020363, Cancer Council of Victoria (CCV) project grant 1052309 and a CCV Venture Grant. N.M is a Sir Henry Dale Fellow, jointly funded by the Wellcome Trust and the Royal Society (Grant Number 211179/Z/18/Z), and also receives funding from Cancer Research UK Lung Cancer Centre of Excellence, Rosetrees, and the NIHR BRC at University College London Hospitals. F.B. receives grants from the European Research Commission (project BCM-UPS, grant #682473) and the Deutsche Forschungsgemeinschaft (SFB 1335 and SFB1243).

## Author contributions

Conceptualisation: P.J.J., N.M., E.M., M.D.; Investigation: E.M., M.D., C.B., D.A., M.J., P.B., M.B., S.U., U.H., A.M., A.W., P.S.P.P., S.L., M.S., X.W.; Resources: M.A., N.P., I.H., R.B., R.M., A.W., T.M., S.F., K.S., J.S.H., M.v.G., F.P., R.R., G.T.S., J.R., F.B., W.W., A.S., M.H., C.S., P.J.J., N.M.; Writing, review, and/or revision of the manuscript: E.M., M.D., C.B., P.J.J., N.M., C.S.; Study supervision: P.J.J., N.M., C.S.

## Funding

## Competing interests

S.F. has had a consulting or advisory role, received honoraria, research funding, and/or travel/accommodation expenses from the following for-profit companies: Bayer, Roche, Amgen, Eli Lilly, PharmaMar, AstraZeneca, and Pfizer. W.W. has attended Advisory Boards and served as speaker for Roche, MSD, BMS, AstraZeneca, Pfizer, Merck, Lilly, Boehringer, Novartis, Takeda, Amgen and Astellas. W.W. receives research funding from Roche, MSD, BMS and Bruker. A.S. is an employee at The Walter and Eliza Hall Institute, which receives milestone payments and royalties from Genentech and AbbVie for the development of Venetoclax. A.S. has collaborations with Servier for the development of MCL-1 inhibitors for cancer therapy. N.P. has had a consulting or advisory role for Novartis. C.S. has had a consulting or advisory role, received honoraria, research funding, and/or travel/accommodation expenses from: Pfizer, AstraZeneca, BMS, Roche Ventana, Boehringer Ingelheim, Novartis, GlaxoSmithKline, MSD, BMS, Celgene, Illumina, Sarah Canon Research Institute, Genentech, Roche Ventana, GRAIL, Medicxi, and Dynamo Therapeutics. C.S. has stock options in Apogen Biotechnologies, Epic Bioscience, GRAIL. C.S. is co-founder and has stock options in Achilles Therapeutics. N.M. has stock options and has received consultancy fees from Achilles Therapeutics. P.J.J. has had a consulting or advisory role, received honoraria, research funding, and/or

travel/accommodation expenses from: Abbvie, BMS, Boehringer, Novartis, Pfizer, Servier, and Celgene. All the other authors disclose no potential conflicts of interest.

## Additional information

Enkhtsetseg Munkhbaatar [1,31], Michelle Dietzen [2,3,4,31], Deepti Agrawal [1], Martina Anton [5], Moritz Jesinghaus [6], Melanie Boxberg [6], Nicole Pfarr [6], Pidassa Bidola [7], Sebastian Uhrig [8,9], Ulrike Höckendorf [1], Anna-Lena Meinhardt [1], Adam Wahida [1], Irina Heid [10], Rickmer Braren [10], Ritu Mishra [11], Arne Warth [12,29], Thomas Muley [13,14], Patrina S. P. Poh [15,30], Xin Wang [1], Stefan Fröhling [16,17], Katja Steiger [6,17], Julia Slotta-Huspenina [6,18], Martijn van Griensven [19], Franz Pfeiffer [7], Sebastian Lange [11,20,21], Roland Rad [11,17,20,21], Magda Spella [22], Georgios T. Stathopoulos [23], Jürgen Ruland [17,24], Florian Bassermann [1,11,17], Wilko Weichert [6,17], Andreas Strasser [25,26], Caterina Branca [1], Mathias Heikenwalder [27], Charles Swanton [2,3], Nicholas McGranahan [2,4,32] & Philipp J. Jost [1,11,17,28,32 ✉]

[1]Department of Medicine III, Klinikum rechts der Isar, TUM School of Medicine, Technical University of Munich, Munich, Germany. [2]Cancer Research UK Lung Cancer Center of Excellence, University College London Cancer Institute, Paul O'Gorman Building, London, UK. [3]Cancer Evolution and Genome Instability Laboratory, The Francis Crick Institute, London, UK. [4]Cancer Genome Evolution Research Group, University College London Cancer Institute, University College London, London, UK. [5]Institute of Molecular Immunology and Experimental Oncology, Klinikum rechts der Isar, Technical University of Munich, Munich, Germany. [6]Institute of Pathology, Technical University of Munich, Munich, Germany. [7]Chair of Biomedical Physics, Department of Physics & Munich School of Bioengineering, Technical University of Munich, Garching, Germany. [8]Division of Applied Bioinformatics, German Cancer Research Center, Heidelberg, Germany. [9]Faculty of Biosciences, Heidelberg University, Heidelberg, Germany. [10]Department of diagnostic and interventional radiology, Klinikum rechts der Isar, Technical University of Munich, Munich, Germany. [11]Center for Translational Cancer Research (TranslaTUM), Technical University of Munich, Munich, Germany. [12]Institute of Pathology, University Hospital Heidelberg, Heidelberg, Germany. [13]Translational Research Unit, Thoraxklinik at Heidelberg University, Heidelberg, Germany. [14]Translational Lung Research Centre (TLRC) Heidelberg, member of the German Centre for lung Research (DZL), Heidelberg, Germany. [15]Experimental Trauma Surgery, Department of Trauma Surgery, Klinikum rechts der Isar, Technical University of Munich, Munich, Germany. [16]Department of Translational Medical Oncology, National Center for Tumor Diseases (NCT) Heidelberg and German Cancer Research Center (DKFZ), Heidelberg, Germany. [17]German Cancer Consortium (DKTK), German Cancer Research Center (DKFZ), Heidelberg, Germany. [18]Gewebebank des Klinikums rechts der Isar und der Technischen Universität München Am Institut für Pathologie der TU München, München, Germany. [19]Department cBITE, MERLN Institute, Maastricht University, Maastricht, The Netherlands. [20]Institute of Molecular Oncology and Functional Genomics, TUM School of Medicine, Technical University of Munich, Munich, Germany. [21]Department of Medicine II, Klinikum rechts der Isar, TUM School of Medicine, Technical University of Munich, Munich, Germany. [22]Laboratory for Molecular Respiratory Carcinogenesis, Department of Physiology, Faculty of Medicine, University of Patras, Rio, Greece. [23]Comprehensive Pneumology Center (CPC) and Institute for Lung Biology and Disease (iLBD), Helmholtz Center Munich for Environmental Health, Member of the German Center for Lung Research (DZL), Munich, Germany. [24]Institute of Clinical Chemistry and Pathobiochemistry, School of Medicine, Technical University of Munich, Munich, Germany. [25]The Walter and Eliza Hall Institute of Medical Research, Melbourne, Australia. [26]Department of Medical Biology, The University of Melbourne, Melbourne, Australia. [27]Division of Chronic Inflammation and Cancer, German Cancer Research Center (DKFZ), Heidelberg, Germany. [28]Division of Clinical Oncology, Department of Medicine, Medical University of Graz, Graz, Austria. [29]Present address: Institute of Pathology, Cytopathology and Molecular Pathology UEGP MVZ, Giessen, Wetzlar, Limburg, Germany. [30]Present address: Julius Wolff Institute, Charité - Universitätsmedizin Berlin, Berlin, Germany. [31]These authors contributed equally: Enkhtsetseg Munkhbaatar, Michelle Dietzen. [32]These authors jointly supervised this work: Nicholas McGranahan, Philipp J. Jost. ✉email: philipp.jost@tum.de

