## [Peer Review File · Nature Communications]

Reviewers' comments:

Reviewer #1 (Remarks to the Author): Lung cancer functional genomics and in vivo models

In this manuscript Munkhbaatar et al explore the role of MCL1 amplification in NSCLC and its potential for therapeutic targeting. First, they demonstrate the MCL1 resides within an amplification peak on 1q21 which includes broad and focal gains, and correlate increased copy number with elevated MCL1 mRNA levels in the TRACERx and TCGA datasets. They then demonstrate that MCL1 protein levels are increased in samples with FISH positivity. Next, they use 3 different cell lines with varying genomic backgrounds and MCL1 protein expression to test MCL1 dependency and confirm selective dependency of MCL1 high lines to specific BIM variants and S63845. Finally, they demonstrate that MCL1 knockout impairs KRAS-driven lung tumorigenesis and that S63845 treatment impairs tumor growth in the aggressive KP mouse model.

Overall this is a nice manuscript. Given other published work in this space combining MEK inhibitors with MCL1 inhibitors in KRAS-driven lung cancer (Nangia et al, Cancer Discov 2018), however, several points should be strengthened regarding the role of MCL-1 amplification as a biomarker.

Major points:

1. Regarding Figure 1 - Do the findings in terms of the copy number/mRNA expression correlation extend to the Cancer Cell Line Encyclopedia (CCLE)? Similar copy number/gene expression data should be available for a large number of NSCLC cell lines. Also does MCL1 dependency in the publicly available Broad Institute DepMap portal track with the copy number change, mRNA expression, or both?
2. Figure 2 further strengthens the correlation between MCL1 copy number gain and protein expression - however statistics around the FISH/IHC correlation are not provided in Supplementary Table 2. These should be performed and included, even if not significant potentially due to low numbers. This also seems like an important piece of data to include in the main figures and not relegated to Supplementary info.
3. While the data in Figure 3 are nice what is important here is to also demonstrate that copy number really does correlate with protein levels across a larger panel of cell lines. Based upon the CCLE analysis recommended in point 1, it should be possible to identify cell lines at the extremes of copy number/mRNA expression and prove by immunoblot that protein levels really do track with this. Moreover, correlating response to S63845 treatment across this larger panel of cell lines should be included to increase the strength of their conclusions.
4. Figures 4/5 - if lung tissue is available from KP mice at the end of study then MCL1 IHC could be interesting since one might predict direct demonstration of the subclonal heterogeneity in vehicle treated mice, with elimination of MCL1 positive clones in the S63845 treated mice.

Minor point - 2f is mislabeled in text, should be 2e

Reviewer #2 (Remarks to the Author): Lung cancer genetics

In this manuscript, Munkhbaatar et al described a potential therapeutic target for KRAS-driven NSCLC, a cancer type for which no effective therapy is available so far. They identified amplification of an anti-apoptotic gene, MCL-1, as a frequent somatic event in human NSCLC, and showed that MCL-1 mediated apoptosis evasion could be targeted in a mouse model with KRAS and TP53 mutations.

Major points

1. The authors observed MCL-1 amplification in lung adenocarcinoma (LUAD) samples from the TRACERx dataset, which is consistent with previously observed MCL-1 amplification in pan-cancer studies. The data (Sup. Fig.1) show that there is frequent whole chr1q gain. It would be important to run GISTIC2 in the TRACERx study to show how precise and significant is the MCL1 focal gain event.
2. If there are 316 genes that are amplified in the region and are correlated with increased gene expression, how can the authors be sure that the effect on cancer progression is mainly due to the MCL1 gene? How would the effect on tumor evolution and survival be if other putative cancer driver genes amplified in the same region were analyzed in the same way?
3. The authors reported high frequency of MCL-1 gain in LUAD associated with upregulation of mRNA. However, high level amplification is not as frequent. It would be important to show the correlation with gene expression among high level gain, low level gain and no gain of MCL1. If only the high-level MCL1 gain is functionally oncogenic, then most of the conclusions need to be revisited. It would be important for comparison to show the correlation with gene expression for other major putative cancer driver genes with high/low/no amplification in the same region of MCL1.
4. It would be important to also identify the samples with any loss of function / non-synonymous mutations in other cancer driver genes to understand the context of the reported findings between MCL1 gain and TP53 loss.
5. The authors show more frequent subclonal MCL1 events in samples with TP53 clonal mutations. Can they report the size of the amplified segments for the clonal and subclonal MCL1? Are they focal or broad? This would help clarify the specific role of MCL1 vs. other genes in the same amplified region.
6. The authors show no difference between clonal and subclonal MCL1 events between KRAS mutant and KRAS WT tumors. Can they comment on the rationale of conducting experiments in KRAS-mutant mice?
7. One of the three cell lines are lung squamous cell carcinomas (LUSC). Are similar gains of MCL-1 identified in LUSC? What is the rationale of using a LUSC cell line? Can the results be affected by the different histological subtypes?
8. The authors show a stronger effect of MCL1 inhibitors vs. inhibitors of BCL-genes. But what would be the effect of inhibiting other cancer driver genes amplified in the same region of MCL1?
9. While the cell line data and KRAS/MCL-1 mouse model experiment suggest the role of MCL-1 in promoting tumor cell survival, is the MCL-1 inhibitor effect on delayed tumor growth in KRAS/TP53 mice through promoting apoptosis? Can the authors comment on this?

10. It would be important to know the copy number status of the MCL-1 region in each lung cancer cell line tested and the MCL-1 levels in KRAS/TP53 mouse model used for MCL-1 inhibitor tests.

Minor points

Supplementary Fig. 1. Please clarify how focal vs broad events are defined (left barplot). Also, there seems to be more than 61 samples in the plot?

On page 6 the authors comment on Fig. 2f. Do they mean Fig. 2e?

Reviewer #3 (Remarks to the Author): Lung cancer in vivo models

In this manuscript the authors show that MCL1 is frequently amplified in human NSCLC being more pronounced in tumors that carry P53 lesions. In this regard MCL1 amplification in lung adenocarcinoma aligns with observations made in other solid tumors next to lymphomas in which MCL1 has been found to be involved. The notion that MCL1 is preferentially amplified in p53 mutant tumors might well be the result of the genomic instability facilitated by P53 loss-of-function as the authors also suggest. Whereas the data from human tumors are certainly of interest as resource information the mouse experiments do not really add much. Nangia et al. have recently shown that inhibiting MCL1 in combination with MEK inhibitors is quite effective in inhibiting KRAS mutant cell line-based xenograft models. Their work illustrated that xenografted lung adenocarcinoma cell lines are particularly sensitive to MCL1 inhibitors in combination with small molecules that impair MEK. The in this study described use of a conditional Kras mouse model to assess a contribution of MCL1 is certainly elegant but not executed in such a way that it robustly establishes the oncogenic role of Mcl1 overexpression in this autochthonous model system. Although it does show that MCL1 expression is likely important for NSCLC development it fails to mimic the mutational condition resulting in MCL1 overexpression as observed in human NSCLC carrying amplification of chromosome 1q segments.

Specific comments:

- Lung adenocarcinomas that can be induced in conditional Kras mutant mice after activation of KRAS and concomitant genetic deletion of Mcl1 appear similarly sized as those in the presence of a functional Mcl1 allele, it seems that there are only fewer (fig. 4F). The authors need to show whether the tumors that do grow after Cre-mediated tumor induction have indeed lost expression of Mcl1. In comparable studies in which pro-tumorigenic genes have been deleted in mouse models the arising tumors often have not deleted the conditional driver gene. It would strengthen the argument that MCL1 is critical for tumor growth if indeed Mcl1 appeared not deleted in these tumors.

- Given the notion that MCL1 amplification is often associated with P53 loss in human NSCLC does the Kras;p53 (KP) model also shows dependence on Mcl1 as shown for the Kras model for its initial growth? Furthermore, one would expect that the KP model might show evidence of Mcl1 amplification in progressed tumors. Was this monitored? Was this found to be the case in the

S63845-treated tumors that did overall show modestly impaired growth? Or did this lead to overexpression of other anti-apoptotic genes?

- In human NSCLC with an amplification of MCL1, MCL1 is often amplified together with many other potentially oncogenic driver genes as the authors document. Even though MCL1 is an evident candidate, one wonders how significant its contribution is on top of all the other potential candidates. An NSCLC xenograft model with a large 1q amplicon -as present in a large fraction of the human tumors ((suppl. Fig 1)- would be more suited to show the inhibitory effect of S63845 and the dependence of these tumors on MCL1 expression. In this regard the autochthonous mouse model with a conditional Mcl1 allele is not really mimicking the human condition: Overexpression of MCL1 is not modeled in the mouse model. Many other genes not acting as drivers might show a similar phenotype as Mcl1 depletion in the mouse (in fact any essential gene would).

- The 1q amplicon appears to extend very close to the telomeric end. The boundary seems rather sharp and in fact in suppl. Fig 1 the authors list various tumors with specific amplification of a small region at the telomeric site suggesting that there might be important drivers in this segment. It is not clear from the figures/table what genes are located in this region. More discussion of candidates in specifically this region seems warranted given this sharp boundary.

Reviewers' comments:

Based on the comments received, we have substantially revised our manuscript and have addressed the reviewers' questions with a large array of additional experimental data. Indeed, we have included novel data into all main figures apart from Figure 5 as well as extensive new data in the Supplementary data section. We would like to highlight that the points raised by the reviewers helped us to markedly improve our manuscript by incorporating novel and relevant data.

Please find a point-by-point reply to the reviewers' comments below:

Reviewer #1 (Remarks to the Author): Lung cancer functional genomics and *in vivo* models

In this manuscript Munkhbaatar et al explore the role of MCL1 amplification in NSCLC and its potential for therapeutic targeting. First, they demonstrate the MCL1 resides within an amplification peak on 1q21 which includes broad and focal gains, and correlate increased copy number with elevated MCL1 mRNA levels in the TRACERx and TCGA datasets. They then demonstrate that MCL1 protein levels are increased in samples with FISH positivity. Next, they use 3 different cell lines with varying genomic backgrounds and MCL1 protein expression to test MCL1 dependency and confirm selective dependency of MCL1 high lines to specific BIM variants and S63845. Finally, they demonstrate that MCL1 knockout impairs KRAS-driven lung tumourigenesis and that S63845 treatment impairs tumour growth in the aggressive KP mouse model.

Overall this is a nice manuscript. Given other published work in this space combining MEK inhibitors with MCL1 inhibitors in KRAS-driven lung cancer (Nangia et al, Cancer Discov 2018), however, several points should be strengthened regarding the role of MCL-1 amplification as a biomarker.

Major points:

- 1. Regarding Figure 1 - Do the findings in terms of the copy number/mRNA expression correlation extend to the Cancer Cell Line Encyclopedia (CCLE)? Similar copy number/gene expression data should be available for a large number of NSCLC cell lines. Also does MCL1 dependency in the publicly available Broad Institute DepMap portal track with the copy number change, mRNA expression, or both?**

We thank the reviewer for this relevant question that allowed us to improve our manuscript. To determine whether MCL-1 upregulation represents a common feature in cell lines, we took advantage of publicly available data downloaded from the Cell Lines Project from COSMIC. In line with our data from the TRACERx and TCGA cohorts, MCL-1 was frequently amplified in the entire cohort of LUAD cancer cell lines (n=34), and MCL-1 genomic gains correlated with MCL-1 mRNA levels (we have reported this new data in Supplementary Fig. 6). Furthermore, we extended our cohort of NSCLC cell lines (from 3 to 11) and we further identified a significant correlation between MCL-1 copy number, MCL-1 mRNA, and MCL-1 protein levels. This highly relevant data is now included into Fig. 4a and Supplementary Fig. 7 in the revised version of our manuscript.

- 2. Figure 2 further strengthens the correlation between MCL1 copy number gain and protein expression - however statistics around the FISH/IHC correlation are not provided in Supplementary Table 2. These should be performed and included, even if not significant potentially due to low numbers. This also seems like an important piece of data to include in the main figures and not relegated to Supplementary info.**

According to the reviewer comment, we have now added the statistical analysis of the correlation between MCL-1 FISH and MCL-1 protein expression as Figure 3f. As the reviewer predicted, due to the small sample size no significant correlation was detected. However, we would like to point out that none of the samples showing MCL-1 high- or low- level gain in FISH displayed a weak MCL-1 protein staining. These data are now included into Supplementary Table 4.

- 3. While the data in Figure 3 are nice what is important here is to also demonstrate that copy number really does correlate with protein levels across a larger panel of cell lines. Based upon the CCLE analysis recommended in point 1, it should be possible to identify cell lines at the extremes of copy number/mRNA expression and prove by immunoblot that protein levels really do track with this. Moreover, correlating response to S63845 treatment across this larger panel of cell lines should be included to increase the strength of their conclusions.**

We thank the reviewer for his/her appraisal of the data and acknowledge the question that allowed us to implement the in vitro data (see also answer to comment #1) with some exciting new analyses. To explore this, we used 11 different NSCLC cell lines (reported in Supplementary Table 5) and we detected a significant correlation between MCL-1 copy number, MCL-1 mRNA and MCL-1 protein levels. We have included these data into the main Figure 4a and into Supplementary Fig. 7. Moreover, we correlated the susceptibility to S63845 (a selective MCL-1 inhibitor)¹ with MCL-1 mRNA and MCL-1 protein levels in these lung cancer cell lines and included these data into main Figure 4b-c and Supplementary Table 7.

A representation of the improved figure as shown in the manuscript (Figure 4) is shown below.

Fig. 4 / MCL-1 secures survival of lung cancer cell lines
 (a) MCL-1 copy number variation, MCL-1 mRNA and MCL-1 protein levels in the cell lines used for this study. (b) Relative viability of the indicated cell lines after 72 h treatment with 5 μM S63945, as determined by CellTiter-Glo[®]. Results were normalised to vehicle treated (control) cells. (c) Linear regression between MCL-1 protein levels (AU, arbitrary units) and viability after S63945 treatment (dashed lines 95% confidence interval; $R^2=0.5441$, $p=0.0233$).

- 4. Figures 4/5 - if lung tissue is available from KP mice at the end of study then MCL1 IHC could be interesting since one might predict direct demonstration of the subclonal heterogeneity in vehicle treated mice, with elimination of MCL1 positive clones in the S63845 treated mice.**

We thank the reviewer for this comment. Upon MCL-1 inhibitor treatment, tumours in the treated group were significantly smaller than in the vehicle group (after 14 and 30 days, Fig. 6c,d). Due to the lack of availability of an MCL-1 antibody suitable for IHC in mouse lung, we had to analyse Mcl-1 mRNA levels by *in situ* hybridisation. We did not detect any significant difference between MCL-1 inhibitor treated vs control groups, neither as total quantification nor as clonal/sub-clonal patterns (see new data in Supplementary Fig. 11b,c). This is not unexpected due to the fact that S63845 treatment has been shown to increase MCL-1 protein levels, without affecting Mcl-1 mRNA levels¹. Furthermore, these data could suggest that S63845 treatment does not actively select for specific tumour clones (at least in our experimental setting). However, as a confounding factor, the analysed tissues were harvested after 25 days from the end of the treatment (see experimental design in Fig. 6b). During this time-frame the tumours may have masked any effect of S63845 treatment on Mcl-1 mRNA levels and this time may have been too short for the selection of tumour cells with Mcl-1 gene copy gains, which in human lung cancer may take years.

Minor point - 2f is mislabeled in text, should be 2e

We thank the reviewer for the identification of this error. This mislabelling has been corrected.

Reviewer #2 (Remarks to the Author): Lung cancer genetics

In this manuscript, Munkhbaatar et al described a potential therapeutic target for KRAS-driven NSCLC, a cancer type for which no effective therapy is available so far. They identified amplification of an anti-apoptotic gene, MCL-1, as a frequent somatic event in human NSCLC, and showed that MCL-1 mediated apoptosis evasion could be targeted in a mouse model with KRAS and TP53 mutations.

Major points

- 1. The authors observed MCL-1 amplification in lung adenocarcinoma (LUAD) samples from the TRACERx dataset, which is consistent with previously observed MCL-1 amplification in pan-cancer studies. The data (Sup. Fig.1) show that there is frequent whole chr1q gain. It would be important to run GISTIC2 in the TRACERx study to show how precise and significant is the MCL1 focal gain event.**

Thanks to the reviewer's suggestion, we ran GISTIC2 on the TRACERx LUAD and LUSC cohorts (Supplementary Fig. 3d,e). GISTIC2 identified a significant peak of copy number gains around MCL-1 in LUAD, thus confirming our previous findings ($4.03 = -\log_{10}(qvalue)$). This important new data is now presented in Supplementary Fig. 3d of the revised version of the manuscript.

- 2. If there are 316 genes that are amplified in the region and are correlated with increased gene expression, how can the authors be sure that the effect on cancer progression is mainly due to the MCL1 gene? How would the effect on tumor evolution and survival be if other putative cancer driver genes amplified in the same region were analyzed in the same way?**

We thank the reviewer for raising this critical point. We have modified the discussion of the revised manuscript to highlight that we do not exclude the possible involvement of other putative cancer genes on the 1q21 amplicon in tumour evolution and/or survival. However, our specific aim was to determine whether MCL-1 may represent a promising therapeutic target in LUAD. To this end, we implemented our in vitro data, which showed a significant correlation between MCL-1 levels and susceptibility to MCL-1 inhibition (see new data in Figure 4b,c). Notably, the cell viability after treatment did not correlate with the mRNA levels of any other putative cancer gene identified in the 1q21 amplicon. These data are now reported in Supplementary Table 7 in the revised version of our manuscript.

- 3. The authors reported high frequency of MCL-1 gain in LUAD associated with upregulation of mRNA. However, high level amplification is not as frequent. It would be important to show the correlation with gene expression among high level gain, low level gain and no gain of MCL1. If only the high-level MCL1 gain is functionally oncogenic, then most of the conclusions need to be revisited. It would be important for comparison to show the correlation with gene expression for other major putative cancer driver genes with high/low/no amplification in the same region of MCL1.**

We thank the reviewer for this comment. We analysed both TRACERx and TCGA cohorts as suggested (see below). We detected significant differences across groups (high-level gain, low-level gain, and no gain) only in the TCGA cohort (where we had significantly more samples). However, in the TRACERx dataset, which is smaller and also less powered, we did not observe a significant result.

Revision Figure 1: Correlation between MCL-1 genomic gains and MCL-1 mRNA levels in the TRACERx (a) and TCGA (b) cohorts. Data were analysed by one-sided Wilcoxon tests with Bonferroni correction, p value as reported in figure.

Further analysis suggested that a primary reason for not detecting a significant difference between gain and no gain samples in TRACERx is that even the no gain samples have increased copy number and expression values for MCL-1. However, those samples show a higher ploidy which results in a smaller log₂-ratio and therefore does not hit the threshold to be classified as gained.

Revision Figure 2: Correlation between ploidy and total copy number in TRACERx LUAD samples.

We decided to report the groups as gain and no gain to make Fig. 1 easier to read. Regardless, the results from these analyses for MCL-1 and the results obtained for the other putative cancer genes have been now reported in Supplementary Table 3.

- 4. It would be important to also identify the samples with any loss of function / non-synonymous mutations in other cancer driver genes to understand the context of the reported findings between MCL1 gain and TP53 loss.**

We would like to thank the reviewer and based on this comment, we took further advantage of the TRACERx dataset to perform the same analysis we did for the association of MCL-1 gains with mutations in TP53 also for associations of MCL-1 gains with mutations in KRAS, EGFR, and PIK3CA. We did not detect any significant correlation for associations of MCL-1 gains with mutations in KRAS and EGFR (new data shown Supplementary Figure 5). We were not able to perform such an analysis for mutations in PIK3CA due to the small sample size of clonal mutant PIK3CA (n=3). Moreover, we have now also reported an overview of all the mutations detected in the TRACERx dataset in Supplementary Figure 4a.

- 5. The authors show more frequent subclonal MCL1 events in samples with TP53 clonal mutations. Can they report the size of the amplified segments for the clonal and subclonal MCL1? Are they focal or broad? This would help clarify the specific role of MCL1 vs. other genes in the same amplified region.**

We thank the reviewer for this important comment. We analysed the amplified segment size across MCL-1 clonal / sub-clonal gain in TP53 WT and TP53 mutant groups. We did not detect any significant difference and we have now reported this new analysis in Figure 2d. Furthermore, we added the information about TP53 status and clonality of MCL-1 gains into Supplementary Figure 2.

- 6. The authors show no difference between clonal and subclonal MCL1 events between KRAS mutant and KRAS WT tumors. Can they comment on the rationale of conducting experiments in KRAS-mutant mice?**

We thank the reviewer for raising this point. Our aim was to evaluate the potential of MCL-1 inhibition as a new therapeutic approach for LUAD irrespective of the underlying cancer driver. Given that KRAS mutations represent the most common driver mutation in lung adenocarcinoma (25-30%) and targeted therapies for KRAS are not available yet, we chose this model to examine the therapeutic potential of the MCL-1 inhibitor S63845. Doing so, we were able to use a well-established and robust mouse model that recapitulates human LUAD closely.

- 7. One of the three cell lines are lung squamous cell carcinomas (LUSC). Are similar gains of MCL-1 identified in LUSC? What is the rationale of using a LUSC cell line? Can the results be affected by the different histological subtypes?**

We thank the reviewer for this important question that allowed us to substantially improve our study. Based on this comment, we expanded MCL-1 analysis to both LUAD and LUSC samples from two independent cohorts (TCGA and TRACERx). Across cohorts and histological subtypes, we consistently identified MCL-1 genomic gain and increased MCL-1 mRNA levels. This new and exciting data is now incorporated into main Figure 1 and Supplementary Figure 2-3.

While the in vitro experiments suggested that whenever MCL-1 is highly expressed the cells are vulnerable to its inhibition (Fig. 4b,c), we cannot exclude that differences in histological subtype could have influenced the results. We have included a sentence mentioning this possibility in the results section of our revised manuscript.

- 8. The authors show a stronger effect of MCL1 inhibitors vs. inhibitors of BCL-genes. But what would be the effect of inhibiting other cancer driver genes amplified in the same region of MCL1?**

We would like to refer to our answer to Reviewer #2 point 2, that our prime intention was to analyse the relevance of MCL-1 in lung cancer. We showed that among the proteins encoded by genes present within the 1q21 amplicon, MCL-1 can be targeted and therefore might hold therapeutic potential irrespective of additional genes present on the amplicon. In addition, we would like to mention that to the best of our knowledge no known specific inhibitors are available for any of the proteins encoded by any of the other putative cancer genes identified by our analysis, except for ABL2 (TKIs such as Dasatinib). Yet, our analysis cannot exclude the possible importance of other genes on the 1q21 amplicon in lung cancer progression and survival. We discuss these considerations in greater detail in our revised manuscript.

- 9. While the cell line data and KRAS/MCL-1 mouse model experiment suggest the role of MCL-1 in promoting tumor cell survival, is the MCL-1 inhibitor effect on delayed tumor growth in KRAS/TP53 mice through promoting apoptosis? Can the authors comment on this?**

This is a relevant questions that we approached by analyzing cleaved caspase 3 staining, as marker of apoptosis. However, we did not detect any significant difference between treated and control group (see data in Supplementary Figure 11d-g). We consider this finding to be not unexpected due to the fact that clearance of apoptotic cells/bodies occurs relatively rapid after apoptosis induction by macrophages or alternative immune cells. In our experimental settings, 25 days passed between the end of the treatment and the tissue collection, which exceeds the timing to detect cleaved caspase 3 positive cells in viable tissue (see experimental design in Fig. 6b).

- 10. It would be important to know the copy number status of the MCL-1 region in each lung cancer cell line tested and the MCL-1 levels in KRAS/TP53 mouse model used for MCL-1 inhibitor tests.**

We thank the reviewer for this comment, which was also raised by Reviewer #1 (please refer also to the answer to Reviewer #1 point 1 and point 3) . In order to correlate MCL-1 copy number gain to functional relevance, we expanded our cell line panel to 11 different NSCLC cell lines. Notably, we detected a significant correlation between MCL-1 genomic gains, MCL-1 mRNA and MCL-1 protein levels (Fig. 4a and Supplementary Fig. 7).

Furthermore, we analysed Mcl-1 mRNA levels by in situ hybridisation in lesions from our experimental mice (see new data included into Figure 6a). In accordance with the TRACERx data, we identified higher level of Mcl-1 mRNA in p53 deleted tumour lesions.

Minor points

Supplementary Fig. 1. Please clarify how focal vs broad events are defined (left barplot). Also, there seems to be more than 61 samples in the plot?

We apologise for this confusion. We used a threshold of 20 Mb to distinguish between focal and broad gains (this is now mentioned in the Methods section). For the sample size, we have analysed more than one tumour region per patient. We have now reported an overview of the number of patients and samples in Supplementary Fig. 1 in our revised manuscript.

On page 6 the authors comment on Fig. 2f. Do they mean Fig. 2e?

We thank the Reviewer for spotting this error that has now been fixed.

Reviewer #3 (Remarks to the Author): Lung cancer in vivo models

In this manuscript the authors show that MCL1 is frequently amplified in human NSCLC being more pronounced in tumors that carry P53 lesions. In this regard MCL1 amplification in lung adenocarcinoma aligns with observations made in other solid tumors next to lymphomas in which MCL1 has been found to be involved. The notion that MCL1 is preferentially amplified in p53 mutant tumors might well be the result of the genomic instability facilitated by P53 loss-of-function as the authors also suggest. Whereas the data from human tumors are certainly of interest as resource information the mouse experiments do not really add much. Nangia et al. have recently shown that inhibiting MCL1 in combination with MEK inhibitors is quite effective in inhibiting KRAS mutant cell line-based xenograft models. Their work illustrated that xenografted lung adenocarcinoma cell lines are particularly sensitive to MCL1 inhibitors in combination with small molecules that impair MEK.

The in this study described use of a conditional Kras mouse model to assess a contribution of MCL1 is certainly elegant but not executed in such a way that it robustly establishes the oncogenic role of Mcl1 overexpression in this autochthonous model system. Although it does show that MCL1 expression is likely important for NSCLC development it fails to mimic the mutational condition resulting in MCL1 overexpression as observed in human NSCLC carrying amplification of chromosome 1q segments.

Specific comments:

- 1. Lung adenocarcinomas that can be induced in conditional Kras mutant mice after activation of KRAS and concomitant genetic deletion of Mcl1 appear similarly sized as those in the presence of a functional Mcl1 allele, it seems that there are only fewer (fig. 4F). The authors need to show whether the tumors that do grow after Cre-mediated tumor induction have indeed lost expression of Mcl1. In comparable studies in which pro-tumorigenic genes have been deleted in mouse models the arising tumors often have not deleted the conditional driver gene. It would strengthen the argument that MCL1 is critical for tumor growth if indeed Mcl1 appeared not deleted in these tumors.**

We thank the reviewer for raising this valid point. In our model, mutant Kras activation and Mcl-1 deletion are induced by Cre recombinase simultaneously. Therefore, it is unlikely for tumour cells to escape Mcl-1 deletion (please note that a tumour would not be able to grow in the absence of Cre activity since Kras would not be activated) or to upregulate expression of Mcl-1, as it should have been deleted at the same time as Kras activation. Conversely, it was reported that lymphomas driven by the loss of p53 had been selected against conditional deletion of Mcl-1², because loss of p53 and conditional deletion of Mcl-1 were independent events.

This notion is supported by Grabow et al., Blood 2014, where the authors show that upon constitutive loss of MCL-1 lymphoma development was delayed (no escape from MCL-1 loss), whereas only under conditions of a conditional deletion of MCL-1 after loss of p53 and subsequent lymphoma development tumor cells were able to escape MCL-1 deletion². Specifically, in Grabow et al., Blood, 2014, the deletion of p53 (constitutive at lymphoma onset) and the deletion of MCL-1 (conditional based on Lck-Cre or Rosa26-CreERT2) were introduced at different time points.

Furthermore, at 19-week post-infection with the Cre expressing adenovirus, Mcl-1 expression was significantly reduced in the $Kras^{G12D/+};Mcl-1^{-/-}$ tumour lesions in comparison to $Kras^{G12D/+}$ tumour lesions. However, this quantification does not differentiate between Mcl-1 expression in tumour cells and non-tumour cells (possibly infiltrating immune cells). To overcome this problem and to specifically address the point raised by the Reviewer, we generated enriched populations of primary $Kras^{G12D/+};Mcl-1^{-/-}$ lung tumour cells. Genomic DNA from these cells was analysed by Taqman® assay, with a Mcl-1 probe and a housekeeping probe (Tfrc). This analysis identified low to undetectable levels of Mcl-1 in all the samples analysed, confirming the successful deletion of Mcl-1 by Cre recombinase. These new data are now reported in Supplementary Table 8.

2. Given the notion that MCL1 amplification is often associated with P53 loss in human NSCLC does the $Kras;p53$ (KP) model also shows dependence on Mcl1 as shown for the $Kras$ model for its initial growth?

We focused our analysis on MCL-1 as candidate therapeutic target for LUAD therapy; therefore, (i) we chose the clinically more relevant model of $Kras$ mutation in combination with loss of p53 and (ii) we started the treatment when the $Kras^{G12D/+};p53^{\Delta/\Delta}$ tumours were well established (in line with a previous data³). We can argue that probably a concomitant deletion of Mcl-1 in $Kras^{G12D/+};p53^{\Delta/\Delta}$ mice would not elicit any effects, due to a previous report in B cell lymphoma, in which lymphoma formation could overcome an early deletion of Mcl-1⁴. However, we believe that the suggested experiment must be considered a topic beyond the scope of our current project since this would take more than 12 months to complete.

3. Furthermore, one would expect that the KP model might show evidence of Mcl1 amplification in progressed tumors. Was this monitored? Was this found to be the case in the S63845-treated tumors that did overall show modestly impaired growth? Or did this lead to overexpression of other anti-apoptotic genes?

We thank the Reviewer for this question. We detected an increased expression of Mcl-1 in $Kras^{G12D/+};p53^{\Delta/\Delta}$ tumour lesions in comparison to $Kras^{G12D/+}$ tumour lesions. These data are now reported in Figure 6a of our revised manuscript. Moreover, no differences were detected at the end of the treatment with the MCL-1 inhibitor S63845 for Mcl-1 genomic gains or Mcl-1 expression levels (new data included into Supplementary Figure 11a-c).

It has been previously shown, that loss of a single allele of Mcl-1, which results in a 30-50% reduction in MCL-1 protein expression and recapitulates MCL-1 protein inhibition with a drug, did not cause any compensatory dysregulation of other BCL-2 family members. These data have been produced in haematopoietic cells, which are highly sensitive to MCL-1 loss or inhibition^{5,6}. This is further supported by our in vitro data (Supplementary Figure 8f), where we observe no compensatory upregulation of BCL-2 family proteins in response to the functional inhibition of MCL-1 using the BIM_{2A} mutant. Moreover, we did not detect any compensatory changes in the expression of other BCL-2 family members in our analysis of TCGA LUAD samples. Together, our analysis revealed that MCL-1 expression levels do not correlate with particular patterns of expression of other BCL-2 family members.

- 4. In human NSCLC with an amplification of MCL1, MCL1 is often amplified together with many other potentially oncogenic driver genes as the authors document. Even though MCL1 is an evident candidate, one wonder how significant its contribution is on top of all the other potential candidates. An NSCLC xenograft model with a large 1q amplicon -as present in a large fraction of the human tumors ((suppl. Fig 1)- would be more suited to show the inhibitory effect of S63845 and the dependence of these tumors on MCL1 expression. In this regard the autochthonous mouse model with a conditional Mcl1 allele is not really mimicking the human condition: Overexpression of MCL1 is not modeled in the mouse model. Many other genes not acting as drivers might show a similar phenotype as Mcl1 depletion in the mouse (in fact any essential gene would).**

We acknowledge this question and would like to point to our answer to Reviewer # 2 point 2 concerning the role of alternative cancer-related genes on the amplicon 1q21. We would like to stress that we do not exclude the possible involvement of other putative genes on the 1q21 amplicon in tumour evolution and/or survival in lung cancer. Yet, it was our specific aim to demonstrate that MCL-1 may represent a new therapeutic target in LUAD and that 1q21 amplification results in the upregulation of this potentially valuable target. To this end, we implemented our in vitro data to show a significant correlation between MCL-1 levels and susceptibility to MCL-1 inhibition (see new data in Figure 4b,c). Moreover, in the revised version of the manuscript we took extra care to re-phrase the discussion to highlight this aspect

We acknowledge the point that a better functional representation of the role of MCL-1 amplification and response to MCL-1 inhibition would be advisable. We therefore performed these analyses in NSCLC cell lines. Here we identified a clear correlation between the MCL-1 amplification, MCL-1 protein level, and the response to the MCL-1 inhibitor S63845 (new data in main Figure 4 a-c and Supplementary Figure 7). A xenograft model certainly substantiates the cell line data, yet we consider our $Kras^{G12D/+};p53^{\Delta/\Delta}$ endogenous mouse model more relevant from the translational point of view. Indeed, we find in model that $Kras^{G12D/+};p53^{\Delta/\Delta}$ lesions showed increased Mcl-1 levels reminiscent of the human setting (new data in main Figure 6a).

- 5. The 1q amplicon appears to extend very close to the telomeric end. The boundary seems rather sharp and in fact in suppl. Fig 1 the authors list various tumors with specific amplification of a small region at the telomeric site suggesting that there might be important drivers in this segment. It is not clear from the figures/table what genes are located in this region. More discussion of candidates in specifically this region seems warranted given this sharp boundary.**

We thank the reviewer for this important point that we have now included into our manuscript. We have supplemented the provided list of amplified genes with their corresponding positions within the 1q21 region. We identified 316 amplified genes in LUAD and 276 in LUSC, for which copy number is significantly correlated with increased mRNA levels. Among these genes, we identified 14 previously reported putative cancer genes. These data are now reported in Supplementary Table 2-3.

References

1. Kotschy, A. *et al.* The MCL1 inhibitor S63845 is tolerable and effective in diverse cancer models. *Nature* **538**, 477-482 (2016).
2. Grabow, S., Delbridge, A.R.D., Valente, L.J. & Strasser, A. MCL-1 but not BCL-XL is critical for the development and sustained expansion of thymic lymphoma in p53-deficient mice. *Blood* **124**, 3939 (2014).
3. Jackson, E.L. *et al.* The differential effects of mutant p53 alleles on advanced murine lung cancer. *Cancer Res* **65**, 10280-8 (2005).
4. Grabow, S., Delbridge, A.R., Aubrey, B.J., Vandenberg, C.J. & Strasser, A. Loss of a Single Mcl-1 Allele Inhibits MYC-Driven Lymphomagenesis by Sensitizing Pro-B Cells to Apoptosis. *Cell Rep* **14**, 2337-47 (2016).
5. Brinkmann, K. *et al.* The combination of reduced MCL-1 and standard chemotherapeutics is tolerable in mice. *Cell Death And Differentiation* **24**, 2032 (2017).
6. Carrington, E.M. *et al.* Anti-apoptotic proteins BCL-2, MCL-1 and A1 summate collectively to maintain survival of immune cell populations both in vitro and in vivo. *Cell Death Differ* **24**, 878-888 (2017).

REVIEWERS' COMMENTS:

Reviewer #1 (Remarks to the Author):

The authors have satisfactorily addressed my concerns

Reviewer #2 (Remarks to the Author):

I appreciate the extensive work conducted by the authors to respond to my questions and comments (Reviewer #2). I have no additional comments. I think this manuscript provides an important contribution to the field and deserves publication.

Reviewer #3 (Remarks to the Author):

The authors have overall adequately addressed the comments of the reviewers. Although I do not share their argument given to referee 3 that Cre expression in a cell warrants that all conditional alleles are switched (there are numerous examples in which this does not happen, and recombination does not occur in all conditional alleles especially when there is selective pressure against such event as likely would be the case for Mcl1) the low or lack of expression of MCL1 later on settles is issue.

REVIEWERS' COMMENTS:

Reviewer #1 (Remarks to the Author):

The authors have satisfactorily addressed my concerns.

Reviewer #2 (Remarks to the Author):

I appreciate the extensive work conducted by the authors to respond to my questions and comments (Reviewer #2). I have no additional comments. I think this manuscript provides an important contribution to the field and deserves publication.

Reviewer #3 (Remarks to the Author):

The authors have overall adequately addressed the comments of the reviewers. Although I do not share their argument given to referee 3 that Cre expression in a cell warrants that all conditional alleles are switched (there are numerous examples in which this does not happen, and recombination does not occur in all conditional alleles especially when there is selective pressure against such event as likely would be the case for Mcl1) the low or lack of expression of MCL1 later on settles is issue.

We thank the reviewers for the positive feedbacks given to our revised manuscript. We believe that their constructive critiques helped us producing a deeply improved final version. We appreciate the recognition of our work.